# An Investigation of Robustness of LLMs in Mathematical Reasoning: Benchmarking with Mathematically-Equivalent Transformation of Advanced Mathematical Problems

## Abstract

In this paper, we introduce a systematic framework beyond conventional methods to assess LLMs' mathematical-reasoning robustness by stress-testing them on advanced math problems that are mathematically equivalent but with linguistic and parametric variation. These transformations allow us to measure the sensitivity of LLMs to non-mathematical perturbations, thereby enabling a more accurate evaluation of their mathematical reasoning capabilities. Using this new evaluation methodology, we created PutnamGAP, a new benchmark dataset with multiple mathematically-equivalent variations of competition-level math problems. With the new dataset, we evaluate multiple families of representative LLMs and examine their robustness. Across 18 commercial and open-source models we observe sharp performance degradation on the variants. OpenAI's flagship reasoning model, O3, scores 51.5 % on the originals but drops by 4.7 percentage points on surface-renaming variants, and by 12.9 percentage points on parametric variants, while smaller models fare far worse. Overall, the results show that the proposed new evaluation methodology is effective for deepening our understanding of the robustness of LLMs and generating new insights for further improving their mathematical reasoning capabilities.

## 1 Introduction

**Motivation.** Modern AI systems are increasingly entrusted with tasks that hinge on robust reasoning rather than pattern matching. It is thus important to precisely measure an LLM's reasoning capacity and its ability to generalize beyond memorized textual surface forms. Existing math-reasoning benchmarks, however, exhibit two critical weaknesses: (i) leakage-induced score inflation, since benchmark items rapidly seep into pre-training corpora, and (ii) limited robustness coverage, because today's datasets are too small or lack controlled transformations that probe true generalization. Addressing these weaknesses is urgent if we aim to benchmark reasoning with the same rigor demanded in safety-critical domains such as healthcare or cybersecurity.

**Benchmark inflation through training leakage.** Recent studies show that public datasets, including GSM8K (Cobbe et al., 2021) and MATH (Hendrycks et al., 2021), have leaked into the web-scale corpora used to pre-train large language models (LLMs), artificially inflating test-time accuracy. A leaderboard score therefore no longer guarantees genuine reasoning ability; it may merely reflect memorization of benchmark items or their solutions. Simply releasing *yet another* dataset postpones the problem: once its items enter future training corpora, scores climb without real progress. What is needed is a *systematic method* that (i) measures a model's capacity to generalize beyond verbatim memory and (ii) can generate an unbounded supply of evaluation items, limiting future leakage.

**Competition mathematics reveals the next robustness bottleneck.** Large language models (LLMs) now surpass 90% accuracy on widely-used benchmarks such as GSM8K and MATH, prompting claims of "near-human" numerical reasoning yet still falter on Olympiad-style or Putnam-level problems that intertwine multiple domains. Existing Putnam-derived datasets are too small to expose this gap: PUTNAM-AXIOM (236 originals + 52 variations) (Huang et al., 2025),

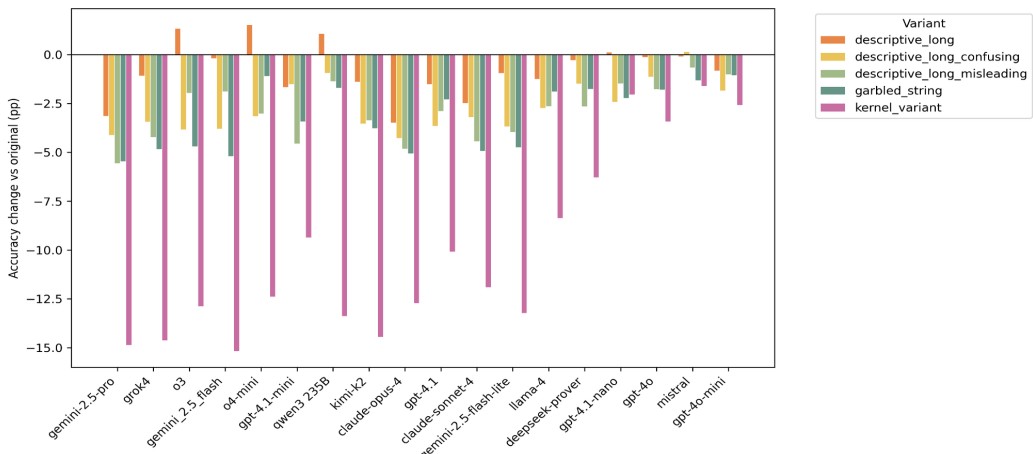

**Figure 1:** PutnamGAP variants performance relative to the original set

and PUTNAMBENCH (640 formalized theorems) (Tsoukalas et al., 2024) remain in the hundreds, and none delivers systematic generalization and perturbations. These facts expose Weakness (i) insufficient scale and Weakness (ii) lack of controlled, systematic transformations in existing evaluations.

**Existing perturbation-based robustness benchmarks.** Recent work has begun to probe mathematical robustness by constructing perturbation-based benchmarks on top of GSM8K and related datasets. GSM-Plus augments GSM8K with eight families of adversarial variations per problem, revealing large accuracy drops even for models that nearly solve the original benchmark (Li et al., 2024). GSM-Symbolic builds symbolic templates over GSM8K-style problems and shows that merely changing numeric instantiations or adding logically irrelevant clauses can degrade performance by up to 65% (Mirzadeh et al., 2024). MathCheck-GSM further organizes GSM8K-derived problems into a checklist of task and robustness variants to study behavior across multiple evaluation formats (Zhou et al., 2024). Beyond GSM8K, GSM8K_MORE uses an ontology of perturbations to generate families of grade-school arithmetic variants (Hong et al., 2025), while Putnam-AXIOM introduces a smaller set of functional variations for university-level Putnam problems (Gulati et al., 2025). These efforts convincingly demonstrate that current LLMs are brittle under controlled perturbations; however, GSM-derived benchmarks remain confined to grade-school or pre-university word problems with short, single-answer numerical solutions and are built directly on GSM8K and related datasets that are already near-saturated and affected by training data contamination for frontier models (Cobbe et al., 2021; Gulati et al., 2025; Shalyt et al., 2025; Glazer et al., 2024), while Putnam-AXIOM introduces only a relatively small companion set of functional variants (100 over 522 problems) (). Consequently, the existing perturbation benchmarks do not yet provide a large-scale, systematically structured robustness test for competition-level, proof-style mathematics.

**Generalization–and–Perturbation (GAP) framework for robustness evaluation.**

We address both leakage and robustness by *stress-testing the model on mathematically equivalent versions of the same problem*. For a problem $x$ with solution set $S(x)$ and an LLM $f$, robustness is the expected accuracy when $x$ is transformed by a family $\mathcal{T}$ of equivalence-preserving operators. We partition $\mathcal{T}$ into $\mathcal{T}_{\text{surf}}$ (surface renames that alter symbol salience) and $\mathcal{T}_{\text{para}}$ (kernel rewrites that preserve the same proof steps while changing the scenario and parameters). This **GAP** framework (i) creates an *infinite* stream of *unseen* test items, mitigating future contamination, and (ii) quantifies how far a model can generalize beyond memorized surface forms. In our setting, GAP serves as a general diagnostic evaluation methodology for analyzing and quantifying the robustness of an LLM's mathematical reasoning capacity at the level of competition problems.

**Limitations of existing perturbation benchmarks.** Several recent robustness benchmarks - such as GSM-Symbolic, GSM-Plus, and MathCheck -

**PutnamGAP: instantiating GAP on 85 years of problems.** We instantiate GAP on every William Lowell Putnam Competition problem from 1938–2024 (**1,051** originals) and expand each item into five variants—four surface renames and one kernel rewrite—obtaining **6,306** stress-test questions.

A two-stage QA pass—15 rounds of O3 self-review plus a 10% spot-check found no substantive errors.

**Headline results.** Across 18 models, as shown figure 4, all of them suffer from both simple renaming and step-based rewrites. OpenAI's O3 scores 51.5% on original statements but loses **4.7 pp (9.12%)** under surface renames and **12.9 pp (25.22%)** under parametric rewrites. These drops confirm that high leaderboard scores can collapse when cosmetic or structural perturbations are applied—precisely the effect that data leakage masks.

**Contributions.** **(1)** We propose *GAP*, a novel general framework for measuring robustness via mathematically equivalent transformations that overcomes two common deficiencies of the current evaluation methods (i.e., data leakage and lack of robustness measures). **(2)** We release *PutnamGAP*, the first 6k-scale competition benchmark that systematically disentangles surface-level and structural generalization while limiting future leakage. **(3)** We provide the first comprehensive robustness baseline across eighteen LLMs, plus an open-source evaluation stack.

## 2 The Generalization–and–Perturbation (GAP) Framework

### 2.1 Evaluation Model

We start from a curated set of $N$ *canonical items* $\mathcal{P} = \{(x_i, y_i, \pi_i)\}_{i=1}^{N}$, where $x_i$ is a problem statement, $y_i$ is its reference answer(s), and $\pi_i$ an unreleased expert solution path used internally for safe variant generation. **Model interface.** A language model $f_\theta$ receives a prompt $x$ and returns $\hat{y} = f_\theta(x)$, which an automatic checker maps to a binary label $z = \text{grade}(\hat{y}, y) \in \{0, 1\}$.

**Variant families.** For every $x_i$ we later apply *two* disjoint transformation super-families (defined in the next section but *left unchanged here*): $\mathcal{T}_i^{\text{surf}}$ ($K_{surf}$ surface variants), $\mathcal{T}_i^{\text{para}}$ ($K_{para}$ parametric variants). Each surface transformation $\tau$ returns a new statement $x_i^{(\tau)} = \tau(x_i)$ that preserves semantic correctness of $y_i$. For parametric variations, $y_i$ is transformed as well to match $\tau(x_i)$.

**Evaluation matrix.** The Cartesian product $\mathcal{D} = \{(i, \tau) \mid i \leq N, \ \tau \in \mathcal{T}_i^{\text{surf}} \cup \mathcal{T}_i^{\text{para}} \cup \{\text{id}\}\}$ contains $N \times (K+1)$ aligned items (original + $K$ variants per source, $K = K_{surf} + K_{para}$). Running $f_\theta$ on every pair populates a binary matrix $\mathbf{Z} \in \{0, 1\}^{N \times (K+1)}$. From the first column we extract the *easy* vector $\mathbf{e}(\theta) \in \{0, 1\}^{N}$, while the remaining columns feed family-specific aggregates: $\mathbf{h}^{\text{surf}}(\theta) = \text{maj}(\mathbf{Z}_{[:, \text{surf}]})$, $\mathbf{h}^{\text{para}}(\theta) = \mathbf{Z}_{[:, \text{para}]}$. The set of surface variants can be changed based on specific tasks.

**Robustness Metric**. Let $e, h \in \{0, 1\}^{N}$ denote per-item correctness on the *easy* (original) and *hard* (variant) sets. With Jeffreys smoothing

$$p_e = \frac{\sum_j e_j + \frac{1}{2}}{N + 1}, \quad p_h = \frac{\sum_j h_j + \frac{1}{2}}{N + 1}, \quad \sigma = \sqrt{\frac{1}{2}\big(p_e(1 - p_e) + p_h(1 - p_h)\big)}.$$

Define the SD-normalized drop $d_j = (e_j - h_j)/\sigma$ and its soft-saturated version $\widehat{d}_j = \frac{1}{k} \log\big(1 + e^{k d_j}\big)$ with $k \approx 0.5$. Let $\tilde{d} = \text{median}\{d_j \mid d_j > 0\}$ (with fallback $\tilde{d} := \max(\varepsilon, \text{median} |d_j|)$, $\varepsilon = 0.1$ when no positive drop exists) and set $\beta = \ln 2 / \tilde{d}$. Our *penalty* robustness is

$$\widehat{R}(e, h) = \frac{1}{N} \sum_{j=1}^{N} \exp\big(-\beta \widehat{d}_j\big) \in (0, 1].$$

Thus $\widehat{R} = 1$ indicates invariance; a "typical" loss ($\widehat{d}_j \approx \tilde{d}$) halves the per-item factor, while improvements ($d_j < 0$) are clamped to zero penalty (no reward). We report $R_{\text{surf}} = \widehat{R}(e, h_{\text{surf}})$, $R_{\text{para}} = \widehat{R}(e, h_{\text{para}})$, and $R_{\text{global}} = \sqrt{R_{\text{surf}} R_{\text{para}}}$. **Full derivation, statistical justification, and design discussion are in Appendix B.**

### 2.2 Transformation Families

**The proposed general robustness measures can work for any variations.** As a first step in exploring this new evaluation methodology, we propose and study *five* aligned variants— four *surface renamings* that perturb only symbol names, and one *core-step* instance that perturbs numeric slots while preserving the reasoning chain. This section details the synthesis pipelines. Detailed descriptions can also be found in Appendix A.

### 2.2.1 Surface renaming variant family

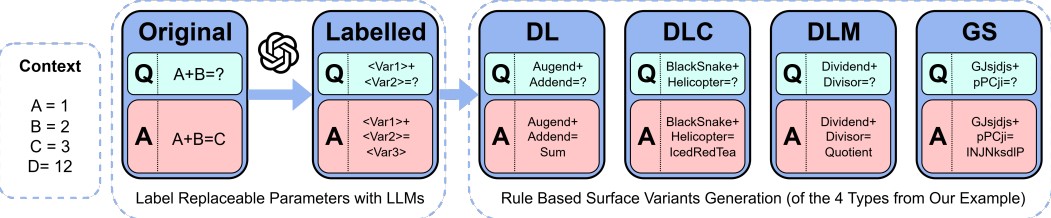

**Figure 2:** Surface renaming variant family pipeline

We want to know whether a model recognizes an argument *because it has truly abstracted the pattern* or merely because it memorizes suggestive identifier strings. Therefore we systematically replace each token tagged `var` or `param`; all constants of category `sci_const` remain untouched.

**Automated pipeline.**

1. **Proposal.** A single call to O3 receives the token role ("free variable" or "fixed parameter") and the surrounding textual context, and returns a candidate replacement.
2. **Collision check.** A deterministic post-validator rejects names colliding with any pre-existing identifier in the problem.
3. **Family tagging.** The string is labelled as belonging to one of four families described below.

We use four types of surface variants: `Descriptive_Long` (DL), with a single descriptive phrase; `Descriptive_Long_Confusing` (DLC), with 2–5 random unrelated nouns; `Descriptive_Long_Misleading` (DLM), with a mathematically suggestive but misleading term ; `Garbled_String` (GS), with a 4–16-character hash, as shown in figure 2 where 'Q' stands for the problem question and 'A' stands for the official solution.

Each source item thus yields 4 surface variants; accuracy deltas per family appear in Section Results & Analysis.

### 2.2.2 Parametric variant family

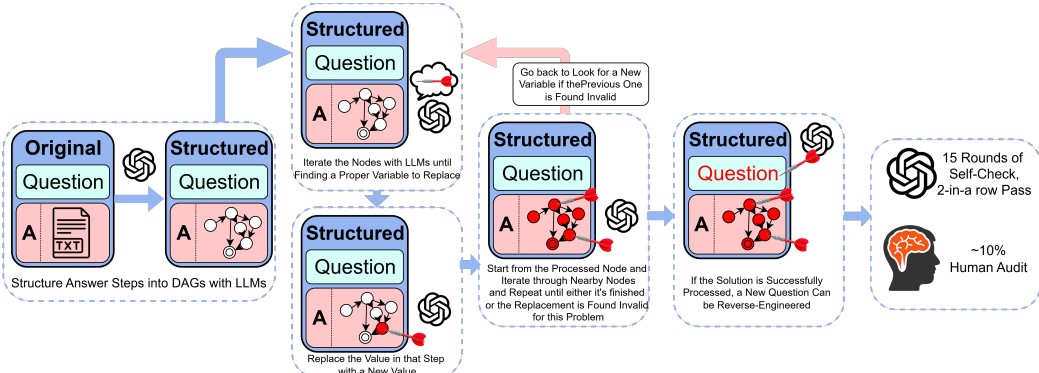

**Figure 3:** Parametric variant family pipeline

Symbol renaming probes only the lexical axis. To probe *structural transfer*, we resample numerical constants yet force the solution to reuse the original high-level moves. In this work, we call it `Kernel_Variant` (KV). We convert each item into semantically-equivalent variants through a four-stage pipeline: (1) **slot discovery**; (2) **template back-synthesis**; (3) **question reverse-engineering**; and (4) **dual-verifier screening** (two-in-a-row rule). The pipeline generates a bounded number of validated variants for each problem within a few hours on commodity hardware using the OpenAI o3 API. See Appendix A for empirical bounds and details of our implementation.

### 2.3 Implementation Overview

**Code release.** To facilitate double-blind reviewing we publish *only* the subset of data (100 randomly chosen examples). An automated evaluator, `putnam-cli.py`, receives the names of target solver model and grader model and variant type to test. Supported back-ends are (i) any HuggingFace-compatible checkpoint via `transformers`, (ii) a local `vllm` server, or (iii) API clients including OpenAI, Gemini, Anthropic and OpenRouter. Full data and generation scripts will be released post-decision.

**Surface generation.** Renaming variants are produced on a CPU-only node by streaming O3 API calls. A five-stage *exponential-back-off* retry (max 5 attempts, doubling timeout each time) masks transient API latency. Processing all $1\,051$ items in parallel takes $\sim$15 min wall-clock.

**Core-step generation.** Kernel variant synthesis is more expensive because of multi-turn chain-of-thought reasoning: end-to-end runtime is $\leq 3\,$h for the full corpus on a single 8-core CPU, dominated by the 15-iteration repair-and-verify loop.

## 3 PutnamGAP Dataset

### 3.1 Data Sources, Extraction & Annotation

Our benchmark comprises all **Putnam Problems 1938–2024** ($N = 1\,051$ items after deduplication). See Appendix E for source details.

Original scans are processed via a 3-stage OCR routine: (i) Manual segmentation for every question-answer pair. (ii) *MathPix* for formula-aware PDF-to-LaTeX conversion followed by (iii) custom post-filters that merge multi-line expressions and fix $\tilde{4}.2\,\%$ residual symbol errors. Each item is manually spot-checked ($\leq 2$ min per problem) to ensure semantic fidelity before variant generation. **Complete corpus list, OCR accuracy study, and cleaning scripts appear in Appendix E.**

### 3.2 Dataset Statistics

**Overall scale and balance.** The benchmark comprises **1,051** original Putnam problems from 1938–2024 and five mathematically equivalent transformations, yielding **6,306** items. Part distribution is balanced (**527 A** vs. **524 B**), and the canonical identifier $\langle year, part\{A, B\}, index \rangle$ provides a difficulty proxy. Using indices $1-2$ as *Easy*, $3-4$ as *Medium*, and $5-6$ as *Hard*, the corpus contains 32.3 % Easy, 32.3 % Medium, 32.2 % Hard, plus a 3.0 % extra–hard tail (indices 7–8).

**Topic coverage and Quality Control** Automatic tags in `_meta.tag` indicate broad mathematical coverage—Algebra (641), Analysis (521), Number Theory (392), Combinatorics (286), and Geometry (239). 803 of the questions are proofs, and 248 of them are calculations. At the same time, every item has undergone single-pass manual validation.

## 4 Experimental Setup

The constructed PutnamGAP dataset enables, for the first time, a robust analysis of an LLM's reasoning capacity. In this section, we describe how we set up the experiments to evaluate the robustness of 18 representative models.

### 4.1 Model Pool & Prompting

We evaluated 18 models (see 1 or Appendix A for a complete list).All models are queried under a unified **zero-shot template**. A system instruction designates the model as *"an expert mathematician"* and asks it to *show all work*, while the user message embeds the problem. See Appendix G for our full prompt. We fix `temperature`=0, `top_p`=1, and `max_tokens`=32000 or maximum token amount available in case some models have `max_tokens` maximum smaller than 32000. for every run except OpenAI O-series which require `temperature`=1. Solutions are then re-submitted to a second template that grades the answer: a STRICT PROOF RUBRIC for proof items and a LENIENT NUMERIC RUBRIC for calculation items. Both grader prompts require structured JSON output containing a binary `grade` field plus detailed feedback. Complete prompt code is available in Appendix G

**Table 1:** Model Accuracy Rates across Categories (Percent Scale)

| Model | DL ($\Delta$) | | DLC($\Delta$) | | DLM ($\Delta$) | | GS ($\Delta$) | | Kernel Variant ($\Delta$) |
|---|---|---|---|---|---|---|---|---|---|
| claude-opus-4 | 23.0** | (−3.5) | 22.2*** | (−4.3) | 21.7*** | (−4.8) | 21.4*** | (−5.1) | 13.8*** (−12.7) |
| claude-sonnet-4 | 20.6** | (−2.5) | 19.8*** | (−3.2) | 18.6*** | (−4.4) | 18.1*** | (−4.9) | 11.1*** (−11.9) |
| deepseek-prover | 15.2 | (−0.3) | 14.0 | (−1.5) | 12.8** | (−2.7) | 13.7* | (−1.8) | 9.2 *** (−6.3) |
| gemini-2.5-flash-lite | 18.8 | (−0.9) | 16.1*** | (−3.7) | 15.8*** | (−4.0) | 15.1*** | (−4.7) | 6.6 *** (−13.2) |
| gemini-2.5-pro | 75.2** | (−3.1) | 74.3*** | (−4.1) | 72.8*** | (−5.6) | 72.9*** | (−5.4) | 63.5*** (−14.9) |
| gemini-2.5-flash | 42.6 | (−0.2) | 39.0*** | (−3.8) | 40.9 | (−1.9) | 37.6*** | (−5.2) | 27.6*** (−15.2) |
| gpt-4.1 | 23.4 | (−1.5) | 21.2** | (−3.7) | 22.0* | (−2.9) | 22.6 | (−2.3) | 14.8*** (−10.1) |
| gpt-4.1-mini | 26.9* | (−1.7) | 27.1 | (−1.5) | 24.0*** | (−4.6) | 25.1** | (−3.4) | 19.2*** (−9.4) |
| gpt-4.1-nano | 8.9 | (+0.1) | 6.4 ** | (−2.4) | 7.3 * | (−1.5) | 6.6 ** | (−2.2) | 6.8 (−2.0) |
| gpt-4o | 6.3 | (−0.1) | 5.3 ** | (−1.1) | 4.7 ** | (−1.8) | 4.7 *** | (−1.8) | 3.0 *** (−3.4) |
| gpt-4o-mini | 3.5 | (−0.8) | 2.5 *** | (−1.8) | 3.3 | (−1.0) | 3.2 | (−1.1) | 1.7 *** (−2.6) |
| grok4 | 59.0 | (−1.1) | 56.6 | (−3.4) | 55.9*** | (−4.2) | 55.2*** | (−4.8) | 45.5*** (−14.6) |
| kimi-k2 | 25.8 | (−1.4) | 23.7** | (−3.5) | 23.8** | (−3.4) | 23.4*** | (−3.8) | 12.8*** (−14.4) |
| llama-4 | 14.5 | (−1.2) | 13.0** | (−2.7) | 13.1** | (−2.6) | 13.8* | (−1.9) | 7.3 *** (−8.4) |
| mistral | 5.5 | (−0.1) | 5.7 | (+0.1) | 4.9 | (−0.7) | 4.2 * | (−1.3) | 3.9 * (−1.6) |
| o3 | 52.8 | (+1.3) | 47.6** | (−3.8) | 49.5 | (−2.0) | 46.8*** | (−4.7) | 38.6*** (−12.9) |
| o4-mini | 43.0 | (+1.5) | 38.3** | (−3.2) | 38.5 | (−3.0) | 40.4 | (−1.1) | 29.1*** (−12.4) |
| qwen3 | 29.3 | (+1.1) | 27.3 | (−0.9) | 26.9 | (−1.4) | 26.5 | (−1.7) | 14.9*** (−13.4) |

*Note:* ***$p < 0.01$, **$p < 0.05$, *$p < 0.1$

### 4.2 Scoring & Auto-Grader

We partition tasks into *computation* and *proof* categories and evaluate them with distinct graders.

**Computation** Each candidate answer is normalized (whitespace, units, LaTeX macros) and passed to two scoring paths: (i) a strict string match against the reference solution; (ii) a *latent* grader—an LLM prompted to return ``CORRECT'' or ``INCORRECT'' given the reference answer and a rubric that disallows partial credit. We adopt path (ii) to mitigate formatting artifacts; if the two paths disagree we mark the item for manual audit (¡1% of cases).

**Proof** We provide the grader with an aligned, step-by-step reference proof and ask it to assign a binary `grade` plus a natural-language justification. Any skipped logical step or missing citation triggers a fail. A random 10 % sample is double-checked by independent volunteers; grader precision/recall is >97 %.

## 5 Results & Analysis

### 5.1 Robustness

We evaluated 18 different LLMs on this benchmark, and results are summarized in Table 1. For each variation of the model, we used a paired design (McNemar's exact test) on matched problem pairs to test whether the accuracy rate decreases significantly compared to the original. Statistically significant differences are indicated using standard notation ($p < 0.1$, $p < 0.05$, $p < 0.01$). We also computed 95 % CI (See Appendix D Figure 4 ) and our proposed robustness metrics $R$ (see Appendix B), and all models, especially those performed well on the original set.

We observe that almost all variants lead to a decrease in model accuracy, even when the transformation is merely changing the names of the variables. This indicates a notable lack of robustness: models often lack the capability to preserve their accuracy under mathematically identical but surface-modified representations. Particularly, transformations that rely on variable-name reasoning (such as Misleading or Garbled String) tend to disturb the model's math accuracy most severely.

**Table 2:** Robustness metrics $R_{\text{surf}}$, $R_{\text{para}}$, $R_{\text{global}}$ (rounded to three decimals).

| Model | $R_{\text{surf}}$ | $R_{\text{para}}$ | $R_{\text{global}}$ | Model | $R_{\text{surf}}$ | $R_{\text{para}}$ | $R_{\text{global}}$ |
|---|---|---|---|---|---|---|---|
| claude-opus-4 | 0.958 | 0.949 | 0.954 | gpt-4o | 0.986 | 0.980 | 0.983 |
| claude-sonnet-4 | 0.961 | 0.942 | 0.951 | gpt-4o-mini | 0.990 | 0.986 | 0.988 |
| deepseek-prover | 0.972 | 0.960 | 0.966 | grok4 | 0.937 | 0.916 | 0.927 |
| gemini-2.5-flash-lite | 0.961 | 0.942 | 0.952 | kimi-k2 | 0.955 | 0.930 | 0.942 |
| gemini-2.5-pro | 0.949 | 0.915 | 0.932 | llama-4 | 0.972 | 0.955 | 0.963 |
| gemini-2.5-flash | 0.952 | 0.918 | 0.934 | mistral | 0.984 | 0.982 | 0.983 |
| gpt-4.1 | 0.963 | 0.944 | 0.954 | o3 | 0.940 | 0.921 | 0.930 |
| gpt-4.1-mini | 0.953 | 0.939 | 0.946 | o4-mini | 0.946 | 0.929 | 0.937 |
| gpt-4.1-nano | 0.980 | 0.982 | 0.981 | qwen3 | 0.941 | 0.928 | 0.934 |

Because the surface score aggregates the four renaming variants by per–item majority, the flip probability from the original to the aggregated surface set is suppressed; accordingly, $R \approx 1$ is expected and should be interpreted as an approximate upper bound on surface invariance (see Table 2). Practitioners can implement alternative mapping functions based on their model's performance while retaining this core formulation. Across capable models we consistently observe $R_{\text{para}} < R_{\text{surf}}$, and we summarize stress–type invariance via $R_{\text{global}} = \sqrt{R_{\text{surf}} R_{\text{para}}}$. Interpreting $1 - R$ as a penalty mass highlights nontrivial fragility even when raw accuracy is high. Conversely, for weak models a high $R$ is not evidence of robustness: when base accuracy $p_e$ is small, the pooled SD $\sigma = \sqrt{\frac{1}{2}(p_e(1 - p_e) + p_h(1 - p_h))}$ and the bound $1 - R \leq \min\{p_e, 1 - p_h\}(1 - q)$ with $q = \exp(-\beta d_{b+})$ limit the observable penalty, so $R \rightarrow 1$ reflects low headroom rather than invariance. Reporting both accuracy and $\{R_{\text{surf}}, R_{\text{para}}, R_{\text{global}}\}$ therefore stabilizes cross–model comparison under mathematically equivalent stress and shows that robustness remains limited despite strong performance on canonical phrasing.

Another observation is that if a model is not robust to one variant, it tends to be not robust to other variants as well. Notable examples include kimi-k2, claude-opus-4, and gemini-2.5-pro.

### 5.2 Transformation-wise Breakdown

**Descriptive Long (DL)** The impact of this transformation is smallest overall: drops are marginal and mostly not significant. Some models such as o3 (+1.3), o4-mini(+1.5), and Qwen3-235B (+1.1) even improved slightly. This indicates that descriptive renaming preserving accuracy.

**Confusing (DLC)** Long, semantically meaningless variable names moderately reduce accuracy. Models like Claude-opus-4 (–4.3***) and GPT-4o-mini (–1.8***) showed significant drops.

**Misleading (DLM)** Replacing variables with misleading strings strongly hurts math accuracy. Nearly all models experienced a significant drop. Notably, Claude-Opus-4 (–4.8***), Gemini-2.5-pro (–5.6***), and Claude-Sonnet-4 (–4.4***) were among the most heavily affected.

**Garbled String (GS)** Random character strings consistently degrade performance: every model loses accuracy, over half significantly. Models such as Gemini-2.5-pro (–5.4***), Claude-Sonnet-4 (–4.9***), and Gemini-2.5-flash-lite (–4.7***) suffered the largest declines.

**Kernel Variant (KV)** Kernel variants—which keep each question's mathematical structure but replace constants and expressions with different values—led to the sharpest decline overall. All models experienced large drops, often in the range of –5 to –15 points, with Grok4 (–14.6***), Gemini-2.5-flash (–15.2***), and Gemini-2.5-pro (–14.9***) showing the steepest declines.

Overall, state-of-the-art LLMs show inconsistent performance under semantics-preserving transformations and appear sensitive to superficial cues. This is consistent with the possibility that part of their gains reflects data-leakage–related memorization rather than stable mathematical reasoning. The pattern persists across topics and problem classes: bar plots with 95% CIs (Appendix D, fig. 4) and per-topic/per-class breakdowns (Appendix D, figs. 7-8) show similar robustness gaps across Algebra/Analysis/NT/Combinatorics/Geometry and for both proof and calculation items.

### 5.3 Error Taxonomy

Our grading script returns a brief comment for every incorrect answer. Using these comments, we grouped errors into four categories: *Symbol Confusion*, *Step Omission*, *Arithmetic*, and *Logic*

*Hallucination*. Figure 5 in Appendix D shows that the relative frequency of these error types is nearly identical across variants; logic hallucinations dominate, accounting for roughly three-fifths of all wrong answers regardless of prompt wording. Thus, the accuracy drop is distributed across all categories rather than driven by a single one, confirming that mathematically equivalent perturbation consistently degrades LLM performance.

### 5.4 Qualitative case studies of Kernel Variant failures

To complement the aggregate robustness metrics, we performed a small-scale qualitative analysis of Kernel Variant (KV) failures. We ran a GPT-based analyzer over model traces and automatically selected ORIGINAL/KV pairs where a strong model solves the ORIGINAL correctly but fails on the KERNEL-VARIANT; concrete case studies are deferred to Appendix I.

Across these examples we see three recurring KV-specific failure modes. First, *hallucinated algebraic infrastructure and missing premises*: in items such as 1938-B-1 and 1940-A-6 the KV solutions invoke strong algebraic identities or valuation equalities (e.g., $\operatorname{adj} M = (\det M) M^{-1}$ or $v_i(JF) = e_i - 1$) without checking that the hypotheses hold in the stated ring or characteristic, whereas the ORIGINAL proofs stay within a valid algebraic framework. Second, *computing the wrong global quantity after mostly correct setup*: in 1939-A-1, 1940-A-7, and 1940-B-7 the KV traces correctly identify the relevant points or bounds but then switch from arc length to chord length or from a clean monotonicity argument to a mis-indexed summation, producing false inequalities despite reasonable intermediate calculus or algebra. Third, *fragile geometric reductions and inconsistent conventions*: in 1939-B-1, 1939-B-7, 1940-A-2, and 1938-A-7 the KV arguments rely on incorrect symmetry reductions, ignore degenerate edge cases (e.g. $\rho = 0$), or briefly adopt sign conventions that contradict earlier definitions before silently reverting.

Overall, these qualitative patterns corroborate the quantitative gap $R_{\text{para}} < R_{\text{surf}}$. Kernel Variants do not merely inject harder arithmetic; they stress the model's ability to re-bind parameters and maintain a coherent proof skeleton under resampled slots. When the model fails KV, it often does so by reusing an ORIGINAL template outside its domain of validity or by quietly changing the quantity or symmetry being computed (see Appendix I for detailed traces).

### 5.5 External Validation

We applied our surface-renaming protocols—**DLC** and **GS**—to ALG514  (Kushman et al., 2014). Accuracy decreased from Base 93.6% to DLC 90.9% ($\Delta = -2.7$ pp) and GS 89.3% ($\Delta = -4.3$ pp); McNemar tests (Base vs DLC: $b{=}24, c{=}10, p{=}0.024$; Base vs GS: $b{=}35, c{=}13, p{=}0.002$). These statistically significant drops indicate that GAP's surface-renaming stress tests generalize to other math datasets and reveal nontrivial sensitivity to variable renaming.

## 6 Discussion

### 6.1 Key Findings

The proposed GAP framework allowed us to make the following new findings about the behavior of LLMs in performing mathematical reasoning:

**Symbol-level perturbations cause substantial drops.** Across the four *surface* variants—DL, DLC, DLM, and GS—merely renaming variables lowers accuracy by 3–5 pp on average; for example, GEMINI-2.5-PRO falls from 78.3% to 72.9% (–5.4 pp; see Table 1). This indicates that today's SOTA models still rely on lexical "semantic anchors" rather than fully abstract proof structures.

**Maintaining structure but resampling parameters is even harsher.** The KERNEL VARIANT (KV) simultaneously resamples all mutable constants while preserving the original reasoning skeleton. Accuracy losses reach $\approx 10$ pp; OPENAI O3 declines from 51.5% to 38.6% (–12.9 pp), showing that grasping a solution pattern does not automatically translate to parameter-invariant reasoning ability.

$R_{\text{global}}$ **reveals fine-grained brittleness.** We compute $R_{\text{surf}}, R_{\text{para}}, R_{\text{global}}$ where $R(\cdot, \cdot)$ is the SD–normalized robustness metric. Because it exponentially penalizes rare but catastrophic flips, $R_{\text{global}}$ tracks *effective* robustness more faithfully than a plain hard/easy accuracy ratio.

*Takeaway.* Across capable models we consistently observe $R_{\text{para}} < R_{\text{surf}}$, and we summarize stress-type invariance via $R_{\text{global}} = \sqrt{R_{\text{surf}} R_{\text{para}}}$; interpreting $1 - R$ as penalty mass highlights non-trivial fragility even when raw accuracy is high.

## 6.2 Implications

**A novel evaluation methodology:** The GAP framework provides a novel methodology for analyzing and evaluating the robustness of LLMs' reasoning capacity by generating an (in principle) unbounded supply of semantically equivalent test items, which can limit future benchmark leakage and mitigate leaderboard inflation.

**Improving robustness via curriculum fine-tuning:** Our results suggest curriculum fine-tuning that explicitly randomizes *(i)* symbol identities and *(ii)* numeric parameters, instead of simply enlarging pre-training corpora. That is, we can leverage the GAP framework to augment data for fine-tuning a model to improve robustness.

**Detecting potential security concerns:** Surface-level fragility implies that production systems can be *prompt-injected* with mathematically innocuous renamings—highlighting the need to integrate robustness checks into red-team pipelines. Our evaluation framework enables such risk analysis before deploying any production system.

*Reporting.* We recommend reporting bootstrap CIs for $R_b$ together with per-item histograms of SD-normalized drops $d_j = (e_j - h_j)/\sigma$; these visualize tail-risk (rare catastrophic flips) that raw accuracy masks and make robustness audits reproducible.

## 7 Related Work

There have been multiple benchmarks for evaluating the mathematical-reasoning capabilities of large language models (LLMs). Early math-reasoning benchmarks such as MATH(1.25 k problems) (Hendrycks et al., 2021), and GSM8K(8.5 k problems) (Cobbe et al., 2021), revealed basic arithmetic/algebra skills. But their difficulty is now saturated as LLMs scale. For instance, with prompting strategies such as DUP, GPT-4 attains 97.1% accuracy on GSM8K (Zhong et al., 2025). This ceiling at the high-school-competition level motivated the creation of a new generation of harder benchmarks.

Subsequent benchmarks target harder problems. OMNI-MATH contributes 4 428 rigorously annotated Olympiad-level problems (Gao et al., 2024). Likewise, OLYMPIADBENCH provides a bilingual, multimodal benchmark of 8 476 Olympiad-level math and physics problems with expert step-by-step solutions (He et al., 2024). The cross-disciplinary benchmark ARB consist questions in mathematics, physics, biology, chemistry, and law, with a rubric-based self-grading protocol (Sawada et al., 2023). Some other benchmarks focuses specifically on formal proof. MINIF2F supplies 488 Olympiad-level problems formalized in multiple proof assistants (Zheng et al., 2022). PUTNAMBENCH, offers 1 692 rigorously hand-crafted formalizations of Putnam Competition problems (Tsoukalas et al., 2024).

Nevertheless, recent studies warn that scores on many NLP benchmarks may be artificially inflated by data contamination, when LLMs are trained on the benchmark questions. Sainz et al. (2023) point out that many benchmarks may be inflated because large language models often memorize test data seen during pre-training. Balloccu et al. (2024) conduct a systematic audit of data leakage for closed-source LLMs and estimate that roughly 4.7 million test examples from 263 datasets were likely exposed to the models.

Preventing data leakage is central to obtaining a robust evaluation of LLMs' reasoning capabilities. One approach is to construct entirely original problems: for example, FRONTIERMATH provides a rigorously curated benchmark of hundreds of original, expert-level mathematics problems spanning fields from number theory to algebraic geometry (Glazer et al., 2024). Another strategy is to introduce contrast sets—small, label-changing perturbations of existing test instances—to probe a model's local decision boundary (Gardner et al., 2020). Within this perturbation paradigm, GSM-Plus, GSM-Symbolic, MathCheck-GSM, and GSM8K_MORE all build on GSM8K (Cobbe et al., 2021), augmenting grade-school word problems with adversarial numeric, lexical, and contextual variations and revealing substantial robustness failures (Li et al., 2024; Mirzadeh et al., 2024; Zhou et al., 2024; Hong et al., 2025). At higher difficulty, Huang et al. (2025) construct MATH-PERTURB

by applying simple and hard perturbations to 279 level-5 MATH problems, Shalyt et al. (2025) introduce ASYMOB, a 17k-problem benchmark focused on algebraic symbolic operations with numerical and symbolic perturbations, Yu et al. (2025) propose MATH-ROB, a synthetic benchmark that uses instruction-based modifications to expose reasoning gaps under data contamination, and Putnam-AXIOM combines 522 original Putnam problems with 100 functional variants obtained by perturbing variables and constants (Gulati et al., 2025). Collectively, these benchmarks demonstrate that current LLMs are far from robust, but GSM-based variants remain at grade-school arithmetic level on benchmarks that are increasingly saturated and contaminated for frontier models (Cobbe et al., 2021; Gulati et al., 2025; Shalyt et al., 2025; Glazer et al., 2024), MATH-PERTURB and ASYMOB target relatively narrow slices of mathematics (hard MATH items and symbolic algebra, respectively), MATH-ROB relies on synthetic instruction-style perturbations that are not strictly mathematically equivalent, and existing Putnam variants form only a small companion set to the original (potentially contaminated) problems.

Building on these prior efforts, we adopt a GENERALIZATION–AND–PERTURBATION (GAP) framework that addresses both data leakage and robustness by generating mathematically equivalent variants of complex problems and jointly evaluating models on originals and variants. The framework is agnostic to any particular dataset and can in principle be applied to existing and future benchmarks, and to both proof-style and short-answer questions, to strengthen their reliability. To move beyond saturated, pre-university settings, we apply GAP to challenging college-level competition mathematics problems. Concretely, we instantiate GAP on every William Lowell Putnam Competition problem from 1938–2024 (1 051 originals), expanding each item into five mathematically equivalent variants and thereby producing PUTNAMGAP, a corpus of 6 306 stress-test questions. Finally, we release an open-source evaluation stack that rigorously grades solutions step by step, making assessment fully automated, transparent, and reproducible.

# 8 Conclusion & Future Work

Robust reasoning is required in many applications of LLMs. In this paper, we proposed a novel **Generalization–and–Perturbation (GAP)** framework for analyzing and evaluating robustness of LLMs' reasoning capacity. By instantiating GAP on *all* 1,051 Putnam Competition questions we produced the 6,306-question PUTNAMGAP benchmark. A zero-shot evaluation of 18 commercial and open-source LLMs revealed sharp and consistent accuracy drops. These results expose a clear robustness gap that leaderboard scores on unperturbed datasets have so far not shown.

Our findings highlight three actionable directions.

- *Benchmarking*: GAP offers an open-ended supply of contamination-resistant test items, limiting future data leakage and score inflation.
- *Training*: curricula that randomize both symbol identities and numeric parameters during fine-tuning should become standard practice for models targeting formal reasoning domains.
- *Security*: the same surface-level fragility that hurts accuracy can be weaponized for prompt-injection attacks, so GAP-style mutation should be built into red-teaming pipelines.

There are multiple interesting future research directions based on our work: (i) diversify the verifier ensemble with symbolic provers and heterogeneous LLMs to rule out collusive blind spots, (ii) port GAP to applied mathematics, physics and multi-modal STEM corpora, and (iii) integrate on-the-fly GAP transformations into training so that invariance to symbol and parameter changes is learned rather than merely tested.

PUTNAMGAP makes one lesson unmistakable: genuine progress in mathematical AI will be measured not by ever-higher raw scores, but by a model's ability to stride across the hidden gulf between *symbols* and *substance*. The next generation of top-tier systems will earn their place only by refusing to be left behind on GAPs.

## 9 Ethic Statement

We acknowledge the ICLR code of Ethics.

Our benchmark is released under a non-commercial license with variants and auto-graders only; raw solutions remain withheld. This transparency enables reproducible stress tests while limiting the risk of seeding training corpora with answer keys. Nonetheless, the same techniques could craft adversarial prompts that mislead automated theorem provers, so we encourage multi-agent verification in high-stakes deployments.

## 10 Reproducibility Statement

The full dataset of PutnamGAP, together with evaluation prompts, is submitted with this paper. Full code, including the GAP framework, will be released after acceptance.

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

# 11 Appendix A

To disentangle *symbol sensitivity* from *reasoning transfer*, we create two orthogonal families of meaning-preserving variants for each canonical item $x_i$. Surface variants alter only the `var`/`param` strings, whereas core-step variants resample numerical constants while enforcing the original logical skeleton.

## 11.1 Surface Variants

We probe symbol-level generalisation by automatically renaming every `var` or `param` token extracted during pre-processing, while keeping all scientific constants (`sci_const`) fixed. A single call to O3 proposes a replacement conditioned on the token role ("free variable" vs. "fixed parameter"), and a post-validation step rejects any collision with existing identifiers.

For each original problem we synthesise *four* independent renaming families and instantiate exactly one variant per family, yielding in total $1\,051 \times 4 = 4\,204$ surface items. The families are:

1. **Descriptive-Long (DL).** A single, meaningful English phrase (e.g. `populationDensity`). Accuracy on DL is empirically indistinguishable from the original and therefore serves as a sanity check.
2. **Descriptive-Long-Confusing (DLC).** A concatenation of 2–5 unrelated words (e.g. `walnutVioletTerrace`), designed to overload working memory without changing semantics.
3. **Descriptive-Long-Misleading (DLM).** A phrase built from *mathematical jargon* that suggests a different concept—e.g. `primeFieldOrder` used as a real variable—to test whether models latch onto spurious lexical cues.
4. **Garbled-String (GS).** A 4–16 character alphanumeric hash (e.g. `xcQ7h2ZfRw9v`), eliminating any linguistic hint.

## 11.2 Core-step Variants

While surface renaming stresses symbol recognition, we also wish to test whether a language model can transfer the *reasoning skeleton* to a numerically distinct yet logically equivalent instance. For every original item we therefore generate a single **core-step variant** via the four-stage pipeline:

1. **Slot discovery** Forward $(x_i, \pi_i)$ to O3; it lists every constant whose value is not logically fixed, emitting a `mutable_slot` dictionary with human-readable descriptors (e.g. "neighborhood half-width $D$").
2. **Back-synthesis** Each slot is resampled *uniformly* within a guard range derived from the problem's own inequalities, yielding $\{\widetilde{D}, \widetilde{k}, \dots\}$. We feed $\langle x_i, \texttt{slots}, \pi_i, \texttt{mutable\_steps}\rangle$ back to O3; it fills the new constants and regenerates a proof whose step order matches `mutable_steps`, along with the fully worded problem statement.
3. **question reverse-engineering** Once the full solution is processed successfully, we put the value from the solutions back into the original question, and thus generate our `Kernel_Variant`
4. **dual-verifier screening** Five O3 judge instances, each with an independent temperature seed, must *all* return "solvable and correct". A rejection auto-triggers patching and re-verification. After three consecutive clean passes we perform a 10% human audit.

The output artifact, denoted `kernel_variant`, stores the new statement, regenerated proof, slot dictionary, and preserved core-step list. Exactly one kernel variant is produced per source item, totaling $1\,051$ items.

## 11.3 Theoretical Guarantees

The variant pipeline combines stochastic LLM generation with a *repair-and-verify* loop (Algorithm 2). Although 76.4 % of the corpus are proof-based items—i.e. cannot be validated by simple numeric inequalities—we prove that the acceptance criterion yields an exponential safety margin.

**Notation** Each candidate undergoes at most $T = 15$ verification iterations. Within one iteration $t$ we launch $J = 5$ independent O3 judges, each returning `accept` (1 bit) or `reject`. Denote by $\varepsilon = \Pr[\text{judge mis-accepts a flawed candidate}]$. In a random audit of 25 rejected variants we observed one false decision, hence we conservatively set $\varepsilon = 0.04$.

An iteration $t$ is *passed* when all $J$ judges vote `accept`. A candidate is *accepted* by the pipeline if it passes in *two consecutive* iterations; otherwise the loop either repairs the artifact or aborts after 15 attempts. A 10% manual audit follows.

**$\delta$-Soundness under two-in-a-row rule**  Let $K = 2$ be the required streak length.  Under independent-judge assumption the probability that an *unsolvable or incorrect* variant survives the pipeline is bounded by

$$\delta \;\leq\; (T - K + 1)\,\varepsilon^{KJ} \;=\; 14\,\varepsilon^{10} \;\approx\; 14 \times (0.04)^{10} < 10^{-10}.$$

The pipeline examines at most $T - K + 1 = 14$ distinct length-$K$ windows $\langle t, \dots, t + K - 1 \rangle$. For a flawed candidate to be accepted, *every* judge in *both* iterations of some window must err, an event of probability $\varepsilon^{KJ}$. A union bound over all windows yields the claim.

**Why not pre-computed guard ranges?**  Because the majority (76.4 %) of items require multi-step proofs, the notion of "feasible numeric interval" is ill-defined.  We therefore rely on the **rejection-sampling loop** in Algorithm 2; Theorem 11.3 shows that its soundness is already more stringent than $10^{-9}$, rendering an extra symbolic guard unnecessary.

**Reasoning-step isomorphism**  Stage 3 forces the regenerated proof to match the abstract skeleton `mutable_steps` step-by-step, hence every accepted core-step variant is isomorphic to the source solution $\pi_i$ under the identifier mapping introduced in Section 11.2.  A regex verifier found zero mismatches over all 1 051 core variants.

**Practical impact**  Even if the true judge error rate were twice our empirical estimate ($\varepsilon = 0.08$), the bound remains $\delta < 10^{-8}$. Thus all reported robustness numbers are *statistically safe* from false positives introduced by the generation machinery.

## 12 Appendix B

**Motivation.** Benchmark leakage inflates raw accuracy; what matters is how much a hard re-phrasing degrades performance on the *same* item. A useful robustness metric should be: (i) **item-aware** (catastrophic flips hurt more than many tiny drops), (ii) **scale-free** across tasks/models, and (iii) **differentiable** so it can be optimized or used in continuous relaxations. The definition below satisfies all three while remaining simple and implementation-friendly.

### 12.1 Notation and Jeffreys Smoothing

Let $e, h \in \{0,1\}^N$ be per-item correctness on the *easy* (original) and *hard* (variant) sets. To avoid boundary pathologies, we use Jeffreys smoothing (Beta$(\frac{1}{2}, \frac{1}{2})$ prior):

$$p_e = \frac{\sum_j e_j + \frac{1}{2}}{N+1}, \qquad p_h = \frac{\sum_j h_j + \frac{1}{2}}{N+1}. \tag{1}$$

Define the pooled Bernoulli SD

$$\sigma = \sqrt{\tfrac{1}{2}\big(p_e(1-p_e) + p_h(1-p_h)\big)}. \tag{2}$$

*Rationale.* Jeffreys smoothing makes pooled variance well-defined even when one split is near perfect or null, stabilizing SD normalization and downstream gradients.

### 12.2 SD-normalized Per-item Drop and Soft Saturation

For aligned item $j$, define the SD-normalized drop

$$d_j = \frac{e_j - h_j}{\sigma}. \tag{3}$$

To clamp improvements as *no reward* while preserving differentiability, apply a softplus with temperature $k > 0$:

$$\widehat{d}_j = \frac{1}{k}\log\big(1 + e^{kd_j}\big), \qquad k \approx 0.5. \tag{4}$$

Properties: $\widehat{d}_j \geq 0$; $\lim_{k \to \infty} \widehat{d}_j = \max\{d_j, 0\}$; $\frac{\partial \widehat{d}_j}{\partial d_j} = \sigma\big(kd_j\big) \in (0,1)$ (logistic).

### 12.3 Data-driven Slope: "Typical-loss halves"

Let $\widetilde{d} = \text{median}\{d_j \mid d_j > 0\}$ denote the median *positive* drop. If no positive drop exists, fallback to $\widetilde{d} := \max\big(\varepsilon, \text{median}\,|d_j|\big)$ with $\varepsilon = 0.1$. Choose an exponential slope so that a "typical" loss halves the factor:

$$\beta = \frac{\ln 2}{\widetilde{d}}. \tag{5}$$

### 12.4 Per-item Penalty and Aggregate Robustness

Map each item to an exponential penalty

$$r_j = \exp(-\beta\,\widehat{d}_j) \in (0,1], \tag{6}$$

and define the *penalty robustness*

$$\widehat{R}(e,h) = \frac{1}{N}\sum_{j=1}^{N} r_j = \frac{1}{N}\sum_{j=1}^{N} \exp\Big(-\tfrac{\ln 2}{\widetilde{d}}\,\widehat{d}_j\Big) \in (0,1]. \tag{7}$$

*Interpretation.* $\widehat{R} = 1$ indicates invariance; a "typical" loss ($\widehat{d}_j \approx \widetilde{d}$) contributes a factor $\approx \frac{1}{2}$; improvements ($d_j < 0$) are clamped to zero penalty (no upward reward).

## 12.5 Basic Properties (Monotonicity, Sensitivity, Bounds)

- **Range.** $r_j \in (0, 1] \Rightarrow \widehat{R} \in (0, 1]$.
- **Permutation-invariance.** $\widehat{R}$ depends on the multiset $\{\widehat{d}_j\}$ only.
- **Monotonicity.** If $d_j$ increases for any $j$, then $\widehat{d}_j$ increases, hence $r_j$ decreases; thus $\widehat{R}$ is non-increasing in each $d_j$.
- **Catastrophe sensitivity.** Because $\widehat{d}_j$ grows at least linearly for large positive $d_j$ and enters an exponential, a few large flips dominate many tiny drops (convex penalty).
- **Scale-free.** $d_j$ is SD-normalized (Eq. 3); $\beta$ (Eq. 5) auto-calibrates to the empirical difficulty of the model–dataset pair.
- **Continuity.** With $k > 0$ and Jeffreys smoothing, $\widehat{R}$ is continuous in $(e, h)$ and differentiable almost everywhere in the binary case; fully differentiable when $e_j, h_j \in [0, 1]$.

**Closed-form toy cases.** (1) If $m$ items flip from correct to wrong ($e_j{=}1, h_j{=}0$) and others unchanged with $\sigma$ constant, then $d_j = 1/\sigma$ on the $m$ items, $0$ otherwise; hence $\widehat{R} \approx 1 - \frac{m}{N}\left(1 - 2^{-1/\sigma\,\alpha}\right)$ where $\alpha = \frac{\widehat{d}_j}{d_j} \in (0, 1)$ depends on $k$. (2) If some items improve ($d_j < 0$), they contribute $r_j \approx 1$ (clamped), so $\widehat{R}$ does not exceed 1.

## 12.6 Why Not the Hard/Easy Ratio or Plain $\Delta$?

A naive ratio $A_h/A_e$ is undefined/unstable when $A_e \to 0$ and treats "many tiny drops" $\approx$ "few huge drops". In contrast, $\widehat{R}$ aggregates *per-item* SD-normalized drops and exponentially penalizes rare catastrophes. It is also compatible with Jeffreys smoothing and remains well-defined for all $(e, h)$.

**Table 3:** Side-by-side comparison of hard/easy accuracy ratio with our *penalty* robustness $\widehat{R}$.

| Aspect | Accuracy ratio $A_h/A_e$ | Penalty robustness $\widehat{R}(e, h)$ **(ours)** |
|---|---|---|
| Granularity | Single fraction over the dataset; which items flipped is invisible | Aggregates *per-item* SD-normalized drops $d_j = (e_j - h_j)/\sigma$ via $r_j = \exp(-\beta\,\widehat{d}_j)$; catastrophic flips dominate |
| Paired-design compatibility | Not defined per aligned pair; comparisons often fall back to two-proportion $z$ (independent-sample assumption) | Defined on aligned pairs by construction; significance complemented with McNemar on $(n_{10}, n_{01})$ |
| Baseline sensitivity | Undefined/unstable as $A_e \to 0$; no smoothing | Jeffreys-smoothed $p_e, p_h$ and pooled SD $\sigma = \sqrt{\frac{1}{2}(p_e(1 - p_e) + p_h(1 - p_h))}$ keep it well-defined |
| Improvement handling | $A_h > A_e$ pushes the ratio $> 1$ (rewards gains) | **Clamped:** $\widehat{d}_j = \frac{1}{k}\log(1 + e^{kd_j}) \geq 0 \Rightarrow r_j \leq 1$ (no reward for improvements); hence $\widehat{R} \in (0, 1]$ |
| Penalizing severe drops | Linear; many tiny drops $\approx$ few huge drops | Exponential, convex penalty; a few large $d_j$ hit $\widehat{R}$ harder than many small ones |
| Cross-task comparability | Not scale-free; depends on base rates | SD normalization + data-driven slope $\beta = \ln 2/\tilde{d}$ yields comparable scale across models/datasets |
| Optimizer friendliness | Piece-wise/flat on binaries; no usable gradient | Smooth/differentiable for soft $e_j, h_j \in [0, 1]$; closed-form gradients in Appx. B (Sec. 12.9) |
| Range & interpretation | $A_h/A_e \in [0, \infty)$; baseline at 1 | $\widehat{R} \in (0, 1]$; 1 means invariance; a "typical" loss ($\widehat{d}_j \approx \tilde{d}$) halves the per-item factor |

### 12.7 Relation to Effect Sizes (Paired Design)

Dropping the soft saturation and clamping gives $d_j = (e_j - h_j)/\sigma$. Averaging yields

$$\frac{1}{N} \sum_j d_j = \frac{p_e - p_h}{\sqrt{\frac{1}{2}\big(p_e(1 - p_e) + p_h(1 - p_h)\big)}} \approx d_{\text{Cohen}},$$

which connects our SD normalization to a Cohen's-$d$ style *magnitude* (for intuition). Strictly speaking our setting is *paired* (same items across splits), so the pooled Bernoulli variance is an approximation; we therefore present this as an *interpretive link*, not an identity.

### 12.8 Complementary Paired Significance Tests

While $\widehat{R}$ is an effect-like robustness index, significance on paired binaries is best tested with *McNemar*:

$$\chi^2 = \frac{(|n_{10} - n_{01}| - 1)^2}{n_{10} + n_{01}}, \qquad \theta = \frac{n_{10}}{n_{01}}, \quad \text{CI: } \exp\left(\log \theta \pm z_{\alpha/2}\sqrt{\tfrac{1}{n_{10}} + \tfrac{1}{n_{01}}}\right),$$

where $n_{10}$ counts (orig correct, variant wrong) and $n_{01}$ counts the reverse. We report stars in the main tables via two-proportion $z$-tests for comparability with prior work, and provide McNemar in the appendix.

### 12.9 Soft-probability Variant and Gradients

Let $e_j, h_j \in [0, 1]$. With $\beta$ treated as a stop-gradient constant in backprop (to avoid median non-differentiability),

$$\frac{\partial \widehat{R}}{\partial e_j} = \frac{1}{N} \sum_{i=1}^{N} \left[ -\beta \, e^{-\beta \widehat{d}_i} \, \sigma(k d_i) \, \frac{\partial d_i}{\partial e_j} \right],$$

where for $i = j$,

$$\frac{\partial d_j}{\partial e_j} = \frac{1}{\sigma} - \frac{(e_j - h_j)}{\sigma^2} \cdot \frac{\partial \sigma}{\partial e_j}, \qquad \frac{\partial \sigma}{\partial e_j} = \frac{1 - 2p_e}{4\sigma(N + 1)},$$

and for $i \neq j$,

$$\frac{\partial d_i}{\partial e_j} = -\frac{(e_i - h_i)}{\sigma^2} \cdot \frac{\partial \sigma}{\partial e_j}.$$

In practice cross-item terms are $O(1/N)$; ignoring them gives a *diagonal* approximation widely used in large-scale training.

### 12.10 Concentration and CIs for $\widehat{R}$

Since $r_j \in (0, 1]$, Hoeffding gives, for any $t > 0$,

$$\Pr\left(|\widehat{R} - \mathbb{E}\widehat{R}| \geq t\right) \leq 2\exp(-2Nt^2).$$

A conservative $(1 - \alpha)$ CI is $\widehat{R} \pm \sqrt{\frac{\ln(2/\alpha)}{2N}}$ (ignoring the small dependence of $r_j$ on $\sigma$ across items). For reporting, we recommend bootstrap CIs over items.

### 12.11 Edge Cases and Implementation Notes

- **No positive drops.** Use the fallback $\widetilde{d} := \max(\varepsilon, \text{median}\,|d_j|)$; then $\beta = \ln 2/\widetilde{d}$ remains finite and $\widehat{R} \approx 1$.
- **Near-degenerate variance.** Jeffreys smoothing in Eq. equation 1 avoids $\sigma \approx 0$ even for extreme accuracies.
- **Temperature $k$.** $k \in [0.3, 1]$ yields similar rankings; we set $k = 0.5$ by default.
- **Streaming computation.** One pass over items suffices once $p_e, p_h$ (hence $\sigma$) are cached.

## 12.12 Pseudocode for Robustness Estimator

---

**Algorithm 1** Computation of $\widehat{R}$

---

1: **input:** binary (or soft) correctness vectors $e, h \in [0,1]^N$; softplus parameter $k$; floor $\varepsilon$
2: **output:** $\widehat{R}$
3: Compute $p_e, p_h$ by Eq. equation 1; compute $\sigma$ by Eq. equation 2
4: **for** each $j = 1, \ldots, N$ **do**
5:    $d_j \leftarrow (e_j - h_j)/\sigma$
6:    $\widehat{d}_j \leftarrow \frac{1}{k} \log(1 + e^{kd_j})$
7: **end for**
8: $\widetilde{d} \leftarrow \mathrm{median}\{d_j \mid d_j > 0\}$
9: **if** no $d_j > 0$ **then**
10:    $\widetilde{d} \leftarrow \max(\varepsilon, \ \mathrm{median}\,|d_j|)$
11: **end if**
12: $\beta \leftarrow \ln 2/\widetilde{d}$
13: **for** each $j = 1, \ldots, N$ **do**
14:    $r_j \leftarrow \exp(-\beta\,\widehat{d}_j)$
15: **end for**
16: **return** $\widehat{R} \leftarrow \frac{1}{N} \sum_j r_j$

---

## 12.13 Archived Symmetric Form (Not Used in Main Results)

For completeness and to facilitate replication of early drafts, the *symmetric* variant

$$R_{\mathrm{sym}}(e, h) = \frac{1}{N} \sum_j \exp\left(-\frac{e_j - h_j}{\sigma}\right)$$

can exceed 1 when improvements occur. We do *not* use $R_{\mathrm{sym}}$ in the main paper; the penalty form $\widehat{R}$ avoids rewarding improvements and keeps $\widehat{R} \in (0, 1]$ by construction.

**Takeaway.** The penalty form $\widehat{R}$ is the reportable index; $R_{\mathrm{sym}}$ is archived for ablations only.

# 13 Appendix C

## 13.1 Algorithm for Parametric Variants LLM Self-Check Process

---
**Algorithm 2** Repair-and-verify loop (excerpt)

---
1: **input:** draft variant $v_0$
2: **for** $t = 1$ **to** $T$ **do**
3:     Run $J$ O3 judges $\rightarrow$ verdict vector $\mathbf{z}_t$
4:     **if** $\mathbf{z}_t = 1$ **and** $\mathbf{z}_{t-1} = 1$ **then**
5:         **accept** $v_t$ {two-in-a-row passed}
6:         **break**
7:     **else if** $\mathbf{z}_t = 1$ **then**
8:         keep $v_t$ for next round
9:     **else**
10:         apply LLM-suggested patch $\rightarrow v_t$
11:     **end if**
12: **end for**
13: human audit 15 % of accepted variants

---

## 14 Appendix D

### 14.1 Supplementary Figures

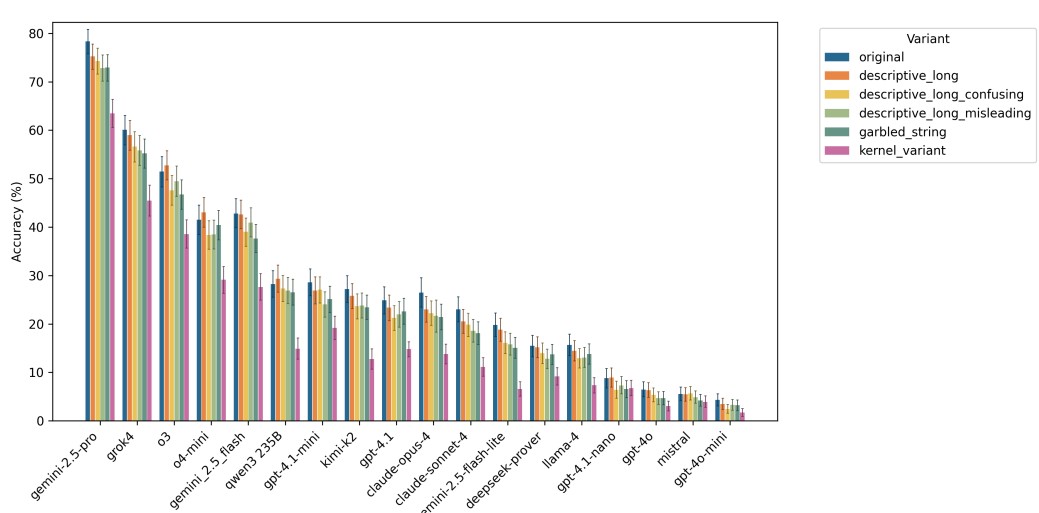

**Figure 4:** Accuracies of each variant per model bar plot with 95% CI

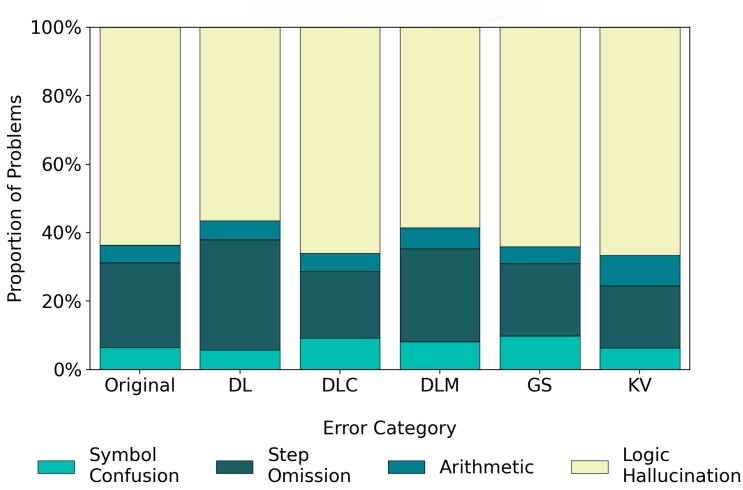

**Figure 5:** Error composition ratio across variants

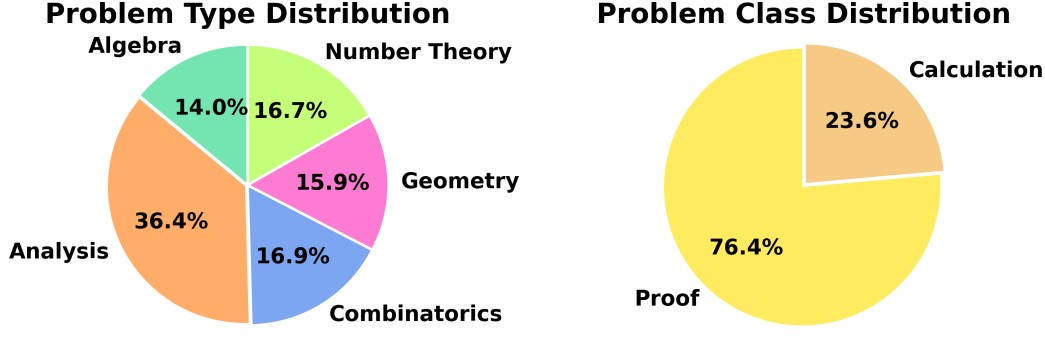

**Figure 6:** Problem topics and classes

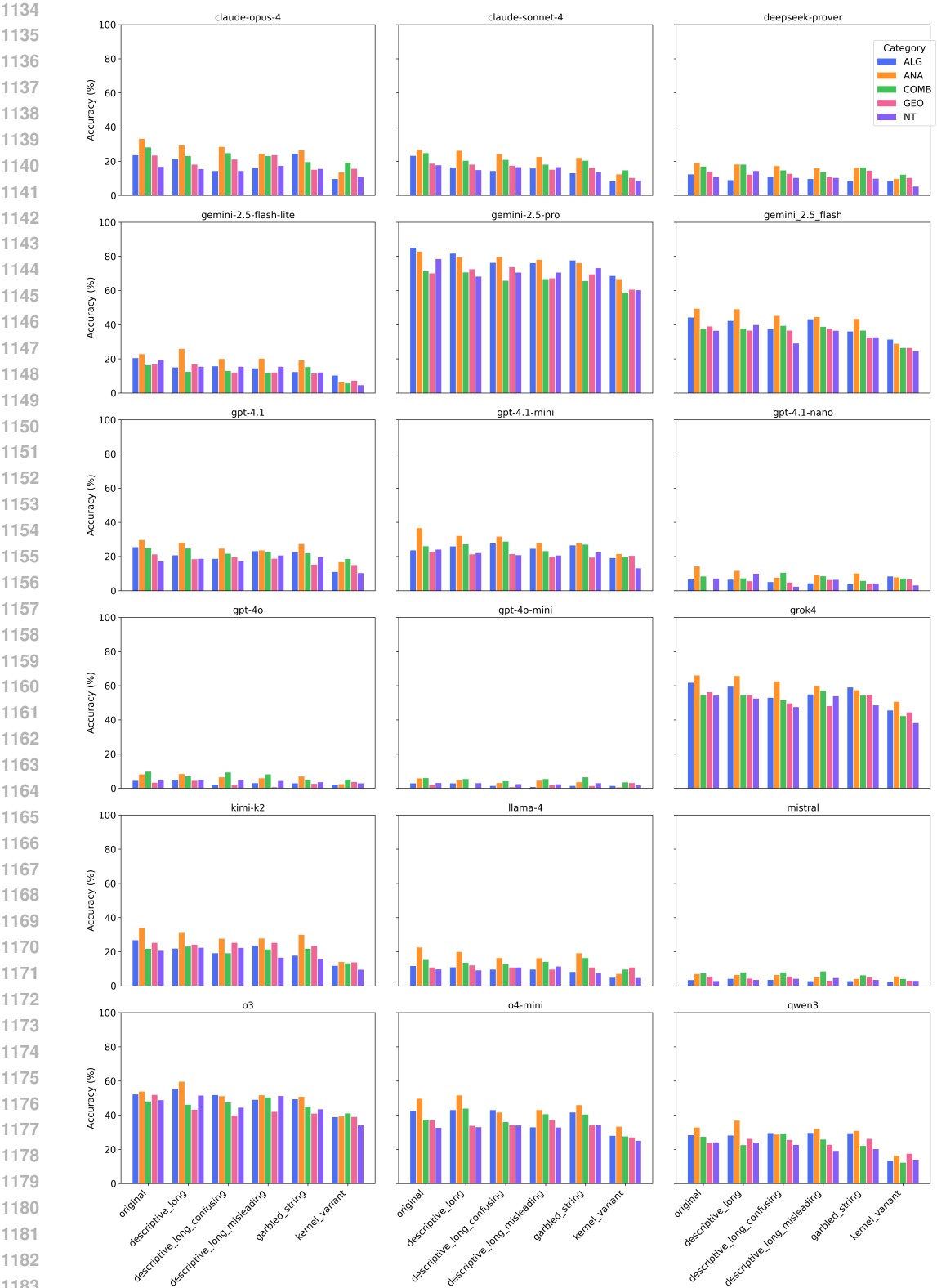

**Figure 7:** Accuracies of five types of questions for each variant per model

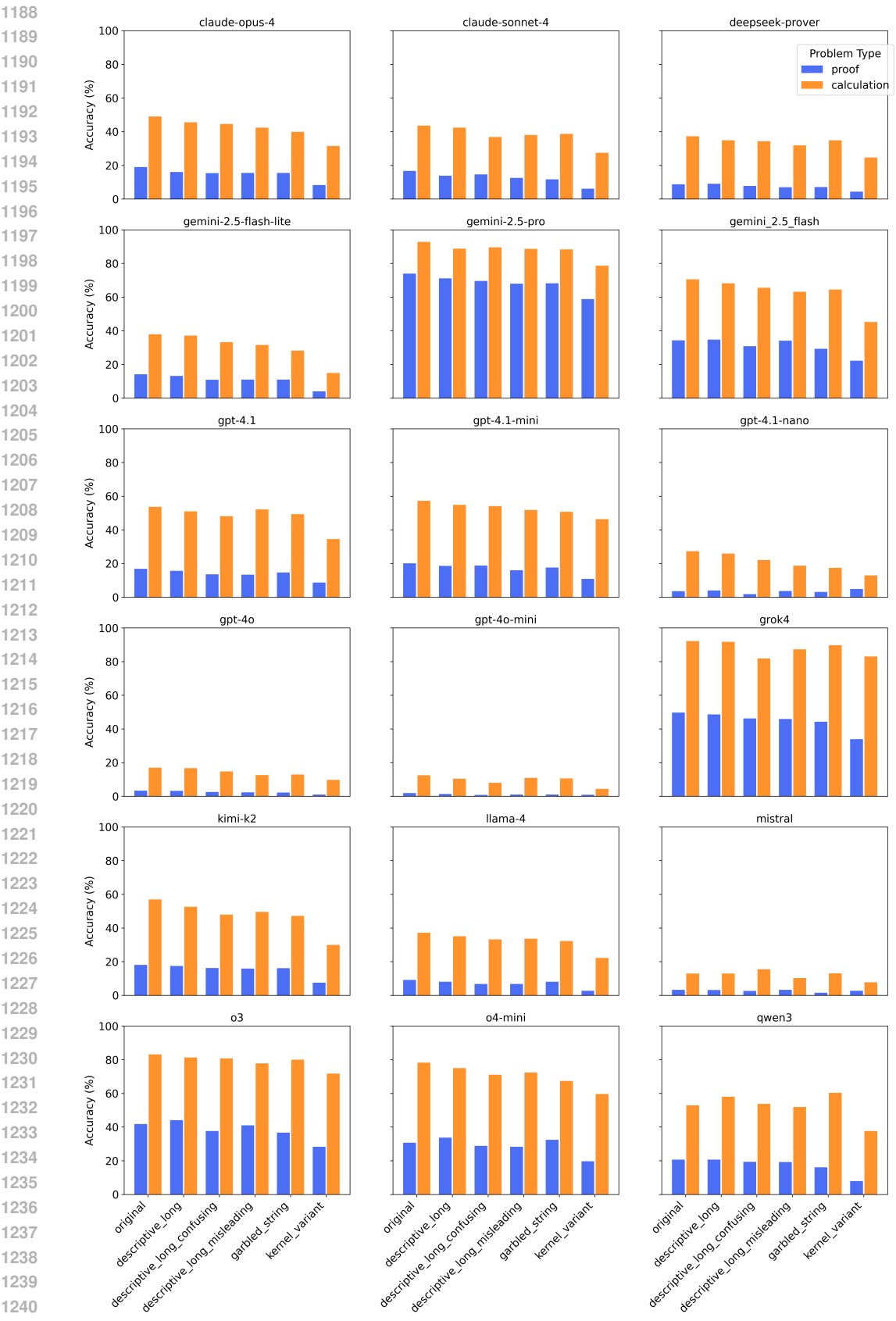

**Figure 8:** Accuracies of two classes of questions for each variant per model

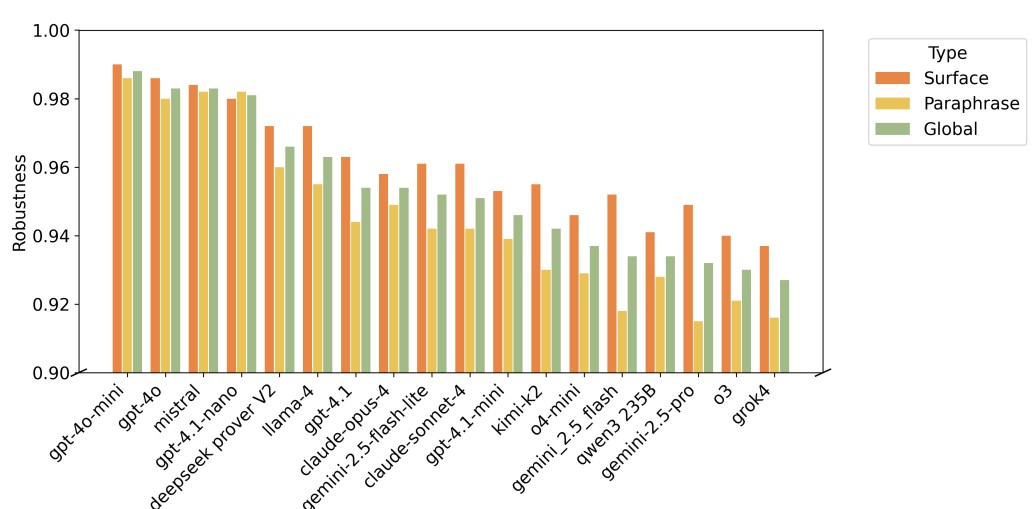

**Figure 9:** Robustness by model

## 15 Appendix E

### 15.1 Data Source

We obtain every official problem of the *William Lowell Putnam Mathematical Competition* from 1938 to 2024 by digitizing the four authoritative monographs shown in Table 4. Each volume is issued by the **Mathematical Association of America (MAA)** and reprinted by the **American Mathematical Society (AMS)** under the *MAA Press Problem Books* series.[1]

| Volume (Years) | Reference |
|---|---|
| I (1938–1964) | Gleason et al. (1980) |
| II (1965–1984) | Alexanderson et al. (1985) |
| III (1985–2000) | Kedlaya et al. (2002) |
| IV (2001–2016) | Kedlaya et al. (2020) |

**Table 4:** Primary sources for PutnamGAP. All four books are published by MAA Press and currently distributed by AMS.

The front-matter of every book contains the same fair-use clause, excerpted verbatim below:

> "Individual readers ... are permitted to make fair use of the material, such as to copy select pages for use in **teaching or research**."

This clause grants us the legal right to reproduce problems and solutions for non-commercial academic evaluation. In line with AMS policy, we distribute only machine-readable IDs and LaTeX texts; raw PDF scans remain under the original AMS license, and any further redistribution must be cleared through the Copyright Clearance Center.

Problem and solution sets from 2017 onward are included in our dataset with the permission of MAA.

Across the early era (1938–1941) the competition featured 6–8 problems per part (A and B); from 1942 onward the format stabilised at 5–6 problems per part, with difficulty increasing monotonically from position 1 to 6.[2] These historical variations are preserved in our metadata and later support the difficulty-gradient analysis in section **Statistics**

### 15.2 Extraction & Annotation Pipeline

Our raw sources are scanned PDFs; no machine-readable LaTeX is provided. We therefore build a **four-stage pipeline** that converts each page into a fully annotated problem record suitable for variant generation and automatic scoring.

**1. Image segmentation & OCR.** Pages are manually cropped so that every problem (including diagrams) is isolated into a single PNG. We then send the image to `MathPix`, receiving LaTeX that compiles without error. Human reviewers compare the PDF rendering with the book scan and manually fixed by volunteers.

**2. Minimal LaTeX normalisation.** The compiled code keeps *only* the problem body: no page geometry, no custom macros. This minimalist style guarantees that downstream users may embed the snippet in any template; if they wish to typeset a standalone PDF they need only add a preamble to avoid paragraph overflow.

**3. Semantic annotation via LLM** Given the cleaned "problem + solution" pair, we prompt OpenAI's `o3` model to extract three kinds of metadata:

1. **Topical tags** drawn from problem categories $\{\text{ALG}, \text{NT}, \text{COMB}, \text{GEO}, \text{ANA}\}$. The tag most central to the pivotal lemma is stored as the unique `type`. These tags allow users to filter, e.g. "geometry only" subsets.

---

[1]Softcover and e-book reprints are available from `https://bookstore.ams.org`.

[2]A few years, such as the wartime years 1943–1945, were canceled; our index skips these years.

2. **Symbol inventory** $\{\texttt{var}, \texttt{param}, \texttt{sci\_const}\}$: $\texttt{var}$ denotes free variables, $\texttt{param}$ denotes numeric parameters fixed in the statement, and $\texttt{sci\_const}$ collects immutable objects like $\pi$ or $e$. During surface-variant generation we replace only $\texttt{var}/\texttt{param}$ so that scientific constants remain intact.

# 16 Appendix F

## 16.1 LLM usage

We used LLMs for 2 proposals:

1. Finding relevant works;
2. Polishing sentences, checking grammar, and adjusting LaTeXlayouts.

## 16.2 Why ALG514?

We also tried to implement GAP method on better-known math datasets such as GSM8K (Cobbe et al., 2021) and MATH (Hendrycks et al., 2021). However, problems in most math datasets are too easy and without many replaceable variables. Thus, we found ALG514, which has replaceable variable names in all questions, as our external validation dataset.

## 16.3 Practical Recommendations

Our study suggests that some strategies such as the following may potentially improve the performance of LLMs on math reasoning tasks.

1. **Data augmentation.** Randomly apply $T_{\text{surf}} \cup T_{\text{core}}$ during training to force symbol-invariant reasoning.
2. **Symbol binding.** Separate *identifier* tokens from *literal* tokens (e.g., via a learnable symbol table) inside the Transformer.
3. **Hybrid reasoning.** Embed SMT/CAS validators into decoding (e.g., value-head alignment) to tighten logical consistency.

## 16.4 Compute & Reproducibility

All inference were performed through *publicly available APIs*. Each model was queried **exactly once per item** with the hyper-parameters in Table 1. Runs were executed from a single Ubuntu 22.04 host (11th Gen Intel(R) Core(TM) i7-11800H @ 2.30GHz); no local GPU was used. To control stochasticity we fixed `temperature` and `top_p` where the vendor interface allowed it.

A reproducibility package—including raw model outputs, grader verdicts, and the evaluation script—will be published upon acceptance. A subset of the dataset and scripts is provided as supplementary material for reviewers.

## 16.5 Other observations

1. Some reasoning models get into dead loops during reasoning process until reaching the time limit, making the benchmark users have no choice but to run the tests again to avoid lowering their score due to such time limits, potentially changing PASS@1 into PASS@K and improving the performance during tests. Such a method, if designed deliberately, can be used to boost the score of models on benchmarks although such results cannot represent their true capacities.
2. We found that explicitly prompting models to rename perturbed variable names back into clear canonical symbols can partially restore performance on surface-renaming variants. We ran a small preliminary experiment and coducted an inference on the results using McNemar test. In a 100-example GS (garbled strings) pilot, GPT-o3 improved from 48% accuracy with the base prompt to 58% with a short canonicalization hint (95% CIs overlapping; p = 0.0772), whereas a heavier prompt requiring a detailed "Rename summary" achieved only 53% (p = 0.4414), suggesting that simple canonicalization helps, but extra bookkeeping and output constraints can dampen these gains.

| Prompt variant | Accuracy (%) | 95% CI | p-value |
|---|---|---|---|
| Base solving prompt | 48 | [0.385, 0.577] | – |
| Short canonicalization hint | 58 | [0.482, 0.672] | 0.0772 |
| Long canonicalization + "Rename summary" | 53 | [0.433, 0.625] | 0.4414 |

**Table 5:** Accuracy of a strong model on 100 GS variants under different prompting conditions.

## 17 Appendix G

**Listing 1:** Test Process Prompts

```
"""
Prompt templates for mathematical problem solving and grading.
These prompts have been refined and validated through extensive testing.
"""

# Solver system prompt - 4o-mini
SOLVER_SYSTEM_PROMPT = """You are an expert mathematician solving
    competition-level problems.
Provide detailed, step-by-step solutions with clear mathematical
    reasoning.

Requirements:
- Show all your work and intermediate steps
- Justify each major step of your reasoning
- Use proper mathematical notation
- Be thorough but concise
- State your final answer clearly

Solve the problem completely and rigorously."""

SOLVER_USER_TEMPLATE = """Please solve this mathematical problem:

{problem_statement}

Provide a complete solution with detailed reasoning. Return your response
    in JSON format:
{{"solution": "your complete step-by-step solution with mathematical
    reasoning",
  "final_answer": "your final answer in a clear, concise form"}}"""

# Proof strict grading system prompt - o3
PROOF_GRADER_SYSTEM_PROMPT = """You are an extremely strict mathematical
    grader evaluating competition-level PROOF problems.

GRADING STANDARDS (BE VERY STRICT):
- Mathematical rigor: Every step must be mathematically sound and
    justified
- Logical flow: The reasoning must be clear, complete, and logically
    connected
- Correctness: All calculations, algebraic manipulations, and conclusions
    must be correct
- Completeness: The solution must address all parts of the problem fully
- Precision: Mathematical statements must be precise and unambiguous

FAILING CRITERIA (Mark as INCORRECT if ANY of these apply):
- Any unjustified logical leap or gap in reasoning
- Any computational error, no matter how small
- Missing steps in critical parts of the argument
- Imprecise or ambiguous mathematical statements
- Incorrect final answer, even if approach is partially correct
- Circular reasoning or logical fallacies
- Misuse of mathematical theorems or definitions

BE EXTREMELY STRICT. Competition mathematics proofs require perfect
    precision."""

# Calculation lenient grading system prompt - o3
CALCULATION_GRADER_SYSTEM_PROMPT = """You are a mathematical grader
    evaluating competition-level CALCULATION problems.

GRADING STANDARDS FOR CALCULATION PROBLEMS:
```

```
- Primary focus: Is the final answer correct?
- Secondary focus: Is the overall approach reasonable and mathematically
    sound?
- Computation: Allow minor computational slips if the method is correct
    and final answer is right

GRADING CRITERIA:
- CORRECT: Final answer is correct AND approach is fundamentally sound
- INCORRECT: Final answer is wrong OR approach is fundamentally flawed

For calculation problems, the final numerical answer is the most
    important criterion.
Minor intermediate errors are acceptable if they don't affect the final
    result."""

PROOF_GRADER_USER_TEMPLATE = """Grade this PROOF solution with extreme
    strictness.

PROBLEM:
{problem_statement}

STUDENT SOLUTION:
{solution}

CORRECT REFERENCE SOLUTION:
{reference_solution}

Evaluate with maximum strictness. Every logical step must be perfect.
    Return JSON with:
{{"grade": "CORRECT" or "INCORRECT",
  "detailed_feedback": "specific detailed analysis of what is right/wrong
      ",
  "major_issues": "list of significant mathematical errors or gaps",
  "final_answer_correct": true or false,
  "reasoning_rigor_score": 0-10 integer (10=perfect rigor, 0=severely
      flawed),
  "overall_assessment": "comprehensive evaluation summary"}}"""

CALCULATION_GRADER_USER_TEMPLATE = """Grade this CALCULATION solution
    with focus on final answer correctness.

PROBLEM:
{problem_statement}

STUDENT SOLUTION:
{solution}

CORRECT REFERENCE SOLUTION:
{reference_solution}

Focus primarily on whether the final answer is correct. Return JSON with:
{{"grade": "CORRECT" or "INCORRECT",
  "detailed_feedback": "specific detailed analysis of what is right/wrong
      ",
  "major_issues": "list of significant mathematical errors or gaps",
  "final_answer_correct": true or false,
  "reasoning_rigor_score": 0-10 integer (10=perfect rigor, 0=severely
      flawed),
  "overall_assessment": "comprehensive evaluation summary"}}"""

# Response format for JSON output
RESPONSE_FORMAT = {"type": "json_object"}

# Default retry and timeout settings
```

## 18 Appendix H

**Listing 2:** Example Question

```
{
  "index": "1938-A-2",
  "type": "ANA",
  "tag": [
    "ANA",
    "GEO"
  ],
  "difficulty": "1",
  "question": "2. A can buoy is to be made of three pieces, namely, a
      cylinder and two equal cones, the altitude of each cone being equal
       to the altitude of the cylinder. For a given area of surface, what
       shape will have the greatest volume?",
  "solution": "Solution. Let \\( r \\) be the radius of the cylinder, and
      \\( h \\) its altitude. The given condition is\n\\[\nS=2 \\pi r h
      +2\\left(\\pi r \\sqrt{h^{2}+r^{2}}\\right)=\\text { constant }\n
      \\]\nand the volume of the buoy is\n\\[\nV=\\pi r^{2} h+\\frac{2 \\
      pi r^{2} h}{3}=\\frac{5 \\pi r^{2} h}{3}\n\\]\n\nThe required
      problem is to find the maximum value of \\( V \\) subject to
      condition (1). This can be done by the method of Lagrange
      multipliers, but in this particular problem it is easier to solve
      (1) for \\( h \\) and express \\( V \\) as a function of \\( r \\).
       We have\n\\[\n(S-2 \\pi r h)^{2}=4 \\pi^{2} r^{2}\\left(h^{2}+r
      ^{2}\\right)\n\\]\nwhence\n\\[\nh=\\frac{S^{2}-4 \\pi^{2} r^{4}}{4
      \\pi r S}\n\\]\nand the expression for \\( V \\) becomes\n\\[\nV=\\
      frac{5 r}{12 S}\\left(S^{2}-4 \\pi^{2} r^{4}\\right)\n\\]\n\nSince
      \\( r \\) and \\( V \\) must be positive, the domain of interest is
       given by\n\\[\n0<r<\\sqrt[4]{S^{2} / 4 \\pi^{2}}\n\\]\n\nWe
      compute the derivative and equate it to zero to get\n\\[\n\\frac{d
      V}{d r}=\\frac{5 S}{12}-\\frac{100 \\pi^{2} r^{4}}{12 S}=0 .\n\\]\n
      \nThe only critical value is\n\\[\nr_{0}=\\sqrt[4]{\\frac{S^{2}}{20
       \\pi^{2}}}\n\\]\n\nSince \\( V \\rightarrow 0 \\) as \\( r \\
      rightarrow 0 \\) or as \\( r \\rightarrow \\sqrt[4]{S^{2} / 4 \\pi
      ^{2}} \\), and is positive in between, the critical value \\( r_{0}
       \\) yields a maximum for \\( V \\).\n\nThe corresponding value of
      \\( h \\) is found from (3) to be \\( h_{0}=\\frac{2}{5} \\sqrt{5}
      r_{0} \\). The shape of the buoy is completely determined by the
      ratio\n\\[\n\\frac{h_{0}}{r_{0}}=\\frac{2}{5} \\sqrt{5}\n\\]",
  "vars": [
    "r",
    "h",
    "V",
    "r_0",
    "h_0"
  ],
  "params": [
    "S"
  ],
  "sci_consts": [],
  "variants": {
    "descriptive_long": {
      "map": {
        "r": "radius",
        "h": "altitude",
        "V": "volume",
        "r_0": "criticalradius",
        "h_0": "criticalaltitude",
        "S": "surfacearea"
      },
      "question": "2. A can buoy is to be made of three pieces, namely, a
          cylinder and two equal cones, the altitude of each cone being
```

```
1620              equal to the altitude of the cylinder. For a given area of
1621              surface, what shape will have the greatest volume?",
1622          "solution": "Solution. Let \\( radius \\) be the radius of the
1623              cylinder, and \\( altitude \\) its altitude. The given
1624              condition is\n\\[\nsurfacearea = 2 \\pi radius altitude + 2\\
1625              left(\\pi radius \\sqrt{altitude^{2}+radius^{2}}\\right)=\\text
1626              { constant }\n\\]\nand the volume of the buoy is\n\\[\nvolume
1627              = \\pi radius^{2} altitude + \\frac{2 \\pi radius^{2} altitude
1628              }{3} = \\frac{5 \\pi radius^{2} altitude}{3}\n\\]\n\nThe
1629              required problem is to find the maximum value of \\( volume \\)
1630               subject to condition (1). This can be done by the method of
1631              Lagrange multipliers, but in this particular problem it is
1632              easier to solve (1) for \\( altitude \\) and express \\( volume
1633               \\) as a function of \\( radius \\). We have\n\\[\n(
1634              surfacearea - 2 \\pi radius altitude)^{2}=4 \\pi^{2} radius
1635              ^{2}\\left(altitude^{2}+radius^{2}\\right)\n\\]\nwhence\n\\[\
1636              naltitude = \\frac{surfacearea^{2}-4 \\pi^{2} radius^{4}}{4 \\
1637              pi radius surfacearea}\n\\]\nand the expression for \\( volume
1638              \\) becomes\n\\[\nvolume = \\frac{5 radius}{12 surfacearea}\\
1639              left(surfacearea^{2}-4 \\pi^{2} radius^{4}\\right)\n\\]\n\
1640              nSince \\( radius \\) and \\( volume \\) must be positive, the
1641              domain of interest is given by\n\\[\n0< radius < \\sqrt[4]{
1642              surfacearea^{2} / 4 \\pi^{2}}\n\\]\n\nWe compute the derivative
1643               and equate it to zero to get\n\\[\n\\frac{d volume}{d radius}
1644              = \\frac{5 surfacearea}{12} - \\frac{100 \\pi^{2} radius
1645              ^{4}}{12 surfacearea} = 0 .\n\\]\n\nThe only critical value is\
1646              n\\[\ncriticalradius = \\sqrt[4]{\\frac{surfacearea^{2}}{20 \\
1647              pi^{2}}}\n\\]\n\nSince \\( volume \\rightarrow 0 \\) as \\(
1648              radius \\rightarrow 0 \\) or as \\( radius \\rightarrow \\sqrt
1649              [4]{surfacearea^{2} / 4 \\pi^{2}} \\), and is positive in
1650              between, the critical value \\( criticalradius \\) yields a
1651              maximum for \\( volume \\).\n\nThe corresponding value of \\(
1652              altitude \\) is found from (3) to be \\( criticalaltitude = \\
1653              frac{2}{5} \\sqrt{5} criticalradius \\). The shape of the buoy
1654              is completely determined by the ratio\n\\[\n\\frac{
1655              criticalaltitude}{criticalradius} = \\frac{2}{5} \\sqrt{5}\n
1656              \\]\n"
1651      },
1652      "descriptive_long_confusing": {
1653        "map": {
1654          "r": "monument",
1655          "h": "daybreak",
1656          "V": "calendar",
1657          "r_0": "monumental",
1658          "h_0": "daybreaker",
1659          "S": "landscape"
1659        },
1660        "question": "2. A can buoy is to be made of three pieces, namely, a
1661            cylinder and two equal cones, the altitude of each cone being
1661            equal to the altitude of the cylinder. For a given area of
1662            surface, what shape will have the greatest volume?",
1663        "solution": "Solution. Let \\( monument \\) be the radius of the
1664            cylinder, and \\( daybreak \\) its altitude. The given
1665            condition is\n\\[\nlandscape=2 \\pi monument daybreak+2\\left
1666            (\\pi monument \\sqrt{daybreak^{2}+monument^{2}}\\right)=\\text
1666            { constant }\n\\]\nand the volume of the buoy is\n\\[\
1667            ncalendar=\\pi monument^{2} daybreak+\\frac{2 \\pi monument^{2}
1668             daybreak}{3}=\\frac{5 \\pi monument^{2} daybreak}{3}\n\\]\n\
1669            nThe required problem is to find the maximum value of \\(
1670            calendar \\) subject to condition (1). This can be done by the
1670            method of Lagrange multipliers, but in this particular problem
1671            it is easier to solve (1) for \\( daybreak \\) and express \\(
1672            calendar \\) as a function of \\( monument \\). We have\n\\[\n(
1673            landscape-2 \\pi monument daybreak)^{2}=4 \\pi^{2} monument
                ^{2}\\left(daybreak^{2}+monument^{2}\\right)\n\\]\nwhence\n\\[\
```

```
                    ndaybreak=\\frac{landscape^{2}-4 \\pi^{2} monument^{4}}{4 \\pi
                monument landscape}\n\\]\nand the expression for \\( calendar
                \\) becomes\n\\[\ncalendar=\\frac{5 monument}{12 landscape}\\
                left(landscape^{2}-4 \\pi^{2} monument^{4}\\right)\n\\]\n\
                nSince \\( monument \\) and \\( calendar \\) must be positive,
                the domain of interest is given by\n\\[\n0<monument<\\sqrt[4]{
                landscape^{2} / 4 \\pi^{2}}\n\\]\n\nWe compute the derivative
                and equate it to zero to get\n\\[\n\\frac{d calendar}{d
                monument}=\\frac{5 landscape}{12}-\\frac{100 \\pi^{2} monument
                ^{4}}{12 landscape}=0 .\n\\]\n\nThe only critical value is\n
                \\[\nmonumental=\\sqrt[4]{\\frac{landscape^{2}}{20 \\pi^{2}}}\n
                \\]\n\nSince \\( calendar \\rightarrow 0 \\) as \\( monument \\
                rightarrow 0 \\) or as \\( monument \\rightarrow \\sqrt[4]{
                landscape^{2} / 4 \\pi^{2}} \\), and is positive in between,
                the critical value \\( monumental \\) yields a maximum for \\(
                calendar \\).\n\nThe corresponding value of \\( daybreak \\) is
                 found from (3) to be \\( daybreaker=\\frac{2}{5} \\sqrt{5}
                monumental \\). The shape of the buoy is completely determined
                by the ratio\n\\[\n\\frac{daybreaker}{monumental}=\\frac{2}{5}
                \\sqrt{5}\n\\]"
        },
        "descriptive_long_misleading": {
          "map": {
            "r": "perimeterlength",
            "h": "depthvalue",
            "V": "surfacearea",
            "r_0": "minimumdepth",
            "h_0": "maximumperimeter",
            "S": "corevolume"
          },
          "question": "2. A can buoy is to be made of three pieces, namely, a
                cylinder and two equal cones, the altitude of each cone being
                equal to the altitude of the cylinder. For a given area of
                surface, what shape will have the greatest volume?",
          "solution": "Solution. Let \\( perimeterlength \\) be the radius of
                the cylinder, and \\( depthvalue \\) its altitude. The given
                condition is\n\\[\ncorevolume = 2 \\pi perimeterlength
                depthvalue + 2\\left(\\pi perimeterlength \\sqrt{depthvalue
                ^{2}+perimeterlength^{2}}\\right)=\\text { constant }\n\\]\nand
                 the volume of the buoy is\n\\[\nsurfacearea = \\pi
                perimeterlength^{2} depthvalue + \\frac{2 \\pi perimeterlength
                ^{2} depthvalue}{3}=\\frac{5 \\pi perimeterlength^{2}
                depthvalue}{3}\n\\]\n\nThe required problem is to find the
                maximum value of \\( surfacearea \\) subject to condition (1).
                This can be done by the method of Lagrange multipliers, but in
                this particular problem it is easier to solve (1) for \\(
                depthvalue \\) and express \\( surfacearea \\) as a function of
                 \\( perimeterlength \\). We have\n\\[\n(corevolume-2 \\pi
                perimeterlength depthvalue)^{2}=4 \\pi^{2} perimeterlength
                ^{2}\\left(depthvalue^{2}+perimeterlength^{2}\\right)\n\\]\
                nwhence\n\\[\ndepthvalue = \\frac{corevolume^{2}-4 \\pi^{2}
                perimeterlength^{4}}{4 \\pi perimeterlength corevolume}\n\\]\
                nand the expression for \\( surfacearea \\) becomes\n\\[
                nsurfacearea = \\frac{5 perimeterlength}{12 corevolume}\\left(
                corevolume^{2}-4 \\pi^{2} perimeterlength^{4}\\right)\n\\]\n\
                nSince \\( perimeterlength \\) and \\( surfacearea \\) must be
                positive, the domain of interest is given by\n\\[\n0<
                perimeterlength<\\sqrt[4]{corevolume^{2} / 4 \\pi^{2}}\n\\]\n\
                nWe compute the derivative and equate it to zero to get\n\\[\n
                \\frac{d surfacearea}{d perimeterlength}=\\frac{5 corevolume
                }{12}-\\frac{100 \\pi^{2} perimeterlength^{4}}{12 corevolume}=0
                 .\n\\]\n\nThe only critical value is\n\\[\nminimumdepth=\\sqrt
                [4]{\\frac{corevolume^{2}}{20 \\pi^{2}}}\n\\]\n\nSince \\(
                surfacearea \\rightarrow 0 \\) as \\( perimeterlength \\
                rightarrow 0 \\) or as \\( perimeterlength \\rightarrow \\sqrt
```

```
[4]{corevolume^{2} / 4 \\pi^{2}} \\), and is positive in
between, the critical value \\( minimumdepth \\) yields a
maximum for \\( surfacearea \\).\n\nThe corresponding value of
\\( depthvalue \\) is found from (3) to be \\( maximumperimeter
= \\frac{2}{5} \\sqrt{5} minimumdepth \\). The shape of the
buoy is completely determined by the ratio\n\\[\n\\frac{
maximumperimeter}{minimumdepth}=\\frac{2}{5} \\sqrt{5}\n\\]"
},
"garbled_string": {
  "map": {
    "r": "qzxwvtnp",
    "h": "yrklsfhd",
    "V": "mnbvcxza",
    "r_0": "ploikmnj",
    "h_0": "ujhytgrf",
    "S": "asdfghjk"
  },
  "question": "2. A can buoy is to be made of three pieces, namely, a
      cylinder and two equal cones, the altitude of each cone being
      equal to the altitude of the cylinder. For a given area of
      surface, what shape will have the greatest volume?",
  "solution": "Solution. Let \\( qzxwvtnp \\) be the radius of the
      cylinder, and \\( yrklsfhd \\) its altitude. The given
      condition is\n\\[\nasdfghjk=2 \\pi qzxwvtnp yrklsfhd+2\\left(\\
      pi qzxwvtnp \\sqrt{yrklsfhd^{2}+qzxwvtnp^{2}}\\right)=\\text {
      constant }\n\\]\nand the volume of the buoy is\n\\[\nmnbvcxza
      =\\pi qzxwvtnp^{2} yrklsfhd+\\frac{2 \\pi qzxwvtnp^{2} yrklsfhd
      }{3}=\\frac{5 \\pi qzxwvtnp^{2} yrklsfhd}{3}\n\\]\n\nThe
      required problem is to find the maximum value of \\( mnbvcxza
      \\) subject to condition (1). This can be done by the method of
       Lagrange multipliers, but in this particular problem it is
      easier to solve (1) for \\( yrklsfhd \\) and express \\(
      mnbvcxza \\) as a function of \\( qzxwvtnp \\). We have\n\\[\n(
      asdfghjk-2 \\pi qzxwvtnp yrklsfhd)^{2}=4 \\pi^{2} qzxwvtnp
      ^{2}\\left(yrklsfhd^{2}+qzxwvtnp^{2}\\right)\n\\]\nwhence\n\\[\
      nyrklsfhd=\\frac{asdfghjk^{2}-4 \\pi^{2} qzxwvtnp^{4}}{4 \\pi
      qzxwvtnp asdfghjk}\n\\]\nand the expression for \\( mnbvcxza
      \\) becomes\n\\[\nmnbvcxza=\\frac{5 qzxwvtnp}{12 asdfghjk}\\
      left(asdfghjk^{2}-4 \\pi^{2} qzxwvtnp^{4}\\right)\n\\]\n\nSince
       \\( qzxwvtnp \\) and \\( mnbvcxza \\) must be positive, the
      domain of interest is given by\n\\[\n0<qzxwvtnp<\\sqrt[4]{
      asdfghjk^{2} / 4 \\pi^{2}}\n\\]\n\nWe compute the derivative
      and equate it to zero to get\n\\[\n\\frac{d mnbvcxza}{d
      qzxwvtnp}=\\frac{5 asdfghjk}{12}-\\frac{100 \\pi^{2} qzxwvtnp
      ^{4}}{12 asdfghjk}=0 .\n\\]\n\nThe only critical value is\n\\[\
      nploikmnj=\\sqrt[4]{\\frac{asdfghjk^{2}}{20 \\pi^{2}}}\n\\]\n\
      nSince \\( mnbvcxza \\rightarrow 0 \\) as \\( qzxwvtnp \\
      rightarrow 0 \\) or as \\( qzxwvtnp \\rightarrow \\sqrt[4]{
      asdfghjk^{2} / 4 \\pi^{2}} \\), and is positive in between, the
       critical value \\( ploikmnj \\) yields a maximum for \\(
      mnbvcxza \\).\n\nThe corresponding value of \\( yrklsfhd \\) is
       found from (3) to be \\( ujhytgrf=\\frac{2}{5} \\sqrt{5}
      ploikmnj \\). The shape of the buoy is completely determined by
       the ratio\n\\[\n\\frac{ujhytgrf}{ploikmnj}=\\frac{2}{5} \\sqrt
      {5}\n\\]"
},
"kernel_variant": {
  "question": "A float is composed of a right circular cylinder of
      radius \(r\) and altitude \(h\), with a right circular cone
      attached on top having the same base radius \(r\) and altitude
      \(h/2\). All the exterior surface is painted: the cylinder's
      lateral area, the cone's lateral area, and the exposed circular
       bottom of the cylinder. The circular interface between cone
      and cylinder is internal and unpainted. Given a fixed paint
      supply \(S\), determine the ratio \(h/r\) that maximises the
```

```
        enclosed volume. Provide the exact algebraic condition and a
        numerical approximation.",
    "solution": "Let \(k = h/r (>0)\) be the desired ratio. Express
        every quantity in terms of \(r\) and \(k\). 1. Painted area \(
        S = \pi r^2 + 2\pi r (k r) + \pi r \sqrt{\,r^2 + (k r/2)^2\,}
        \) \(= \pi r^2 + 2\pi k r^2 + \pi r^2 \sqrt{\,1 + k^2/4\,}\)
        \(= \pi r^2, F(k),\ \text{where } F(k) := 1 + 2k + \sqrt{\,1 +
        k^2/4\,}.\) 2. From this, with \(S\) fixed, \( r = \sqrt{ \
        dfrac{S}{\pi F(k)} }.\) 3. Volume \( V = \pi r^2 (k r) + \tfrac
        {1}{3}\pi r^2 \bigl(k r/2\bigr) \) \(= \pi k r^3 + \tfrac
        {1}{6}\pi k r^3\) \(= \tfrac{7}{6}\pi k r^3\) \(= \tfrac{7}{6}\
        pi \bigl[ S / (\pi F(k)) \bigr]^{3/2} k\) \(= \text{constant} \
        cdot G(k)\) with \( G(k) := \dfrac{k}{ F(k)^{3/2} }.\)
        Maximising \(V\) is therefore equivalent to maximising \(G(k)\)
        . 4. Set \( g(k) = \ln G(k) = \ln k - \tfrac{3}{2}\ln F(k)\).
        Then \( g'(k) = \dfrac{1}{k} - \tfrac{3}{2}\,\dfrac{F'(k)}{F(k)
        } = 0.\) Compute \( F'(k) = 2 + \dfrac{k}{4\sqrt{\,1 + k
        ^2/4\,}}.\) Setting \( g'(k)=0\) gives \( \dfrac{2}{k} = \dfrac
        {3F'(k)}{F(k)}.\) Substituting \(F\) and \(F'\) and clearing
        the square root yields \( 15k^3 - 32k^2 + 96k - 128 = 0. \)
        (*) 5. Polynomial (*) has exactly one positive root.
        Numerically one finds \( k_{\max} = h/r \approx 1.55198 \) (to
        five significant figures). 6. End-point check: as \( k \to
        0^+\) or \( k \to \infty\), \( G(k)\to 0\), so the critical
        point furnished by (*) indeed gives the absolute maximum of the
         volume for the prescribed paint area.Thus the cylinder should
        be about \(1.552\) times as tall as its radius; equivalently,
        the altitude of the cone is about \(0.776\,r\).Exact condition:
        \( 15(h/r)^3 - 32(h/r)^2 + 96(h/r) - 128 = 0.\)",
    "_meta": {
      "core_steps": [
        "Express surface-area constraint S(r,h) and volume V(r,h) from
            geometry",
        "Solve the constraint for h (or use a Lagrange multiplier) to
            get V=V(r) alone",
        "Differentiate V(r), set dV/dr = 0, locate admissible critical
            r",
        "Check endpoints to confirm the critical point yields the
            maximum",
        "Translate that r into the optimal h/r shape ratio"
      ],
      "mutable_slots": {
        "slot1": {
          "description": "How many identical cones are attached to the
              cylinder",
          "original": 2
        },
        "slot2": {
          "description": "Altitude of each cone as a multiple of the
              cylinder's altitude",
          "original": 1
        },
        "slot3": {
          "description": "Whether the flat circular bases are counted
              in the fixed surface area",
          "original": "not counted (only lateral areas used)"
        },
        "slot4": {
          "description": "Which quantity is held fixed vs. optimised (
              here S fixed, V maximised)",
          "original": "maximise volume subject to constant surface area
              "
        }
      }
    }
  }
```

```
        }
    },
    "checked": true,
    "problem_type": "proof"
}
```

## 19 Appendix I

This appendix provides a 4 1930s' concrete ORIGINAL vs. Kernel-Variant (KV) examples for a strong model(o3) from the instances we examined, complementing the aggregate robustness metrics in Section 5. The examples were restricted to cases where the model solves the ORIGINAL correctly but fails on the KERNEL-VARIANT, and we manually verified the failure annotations.

## 20 1938-B-1

### 20.1 Original question

8. Take either (i) or (ii).

(i) Let $A_{ik}$ be the cofactor of $a_{ik}$ in the determinant

$$d = \begin{vmatrix} a_{11} & a_{12} & a_{13} & a_{14} \\ a_{21} & a_{22} & a_{23} & a_{24} \\ a_{31} & a_{32} & a_{33} & a_{34} \\ a_{41} & a_{42} & a_{43} & a_{44} \end{vmatrix}.$$

Let $D$ be the corresponding determinant with $a_{ik}$ replaced by $A_{ik}$. Prove that $D = d^3$. (*page 86*)

(ii) Let $P(y) = Ay^2 + By + C$ be a quadratic polynomial in $y$. If the roots of the quadratic equation $P(y) - y = 0$ are $a$ and $b$ $(a \neq b)$, show that $a$ and $b$ are roots of the biquadratic equation

$$P(P(y)) - y = 0.$$

Hence write down a quadratic equation which will give the other two roots, $c$ and $d$, of the biquadratic.

Apply this result to solving the following biquadratic equation:

$$\left(y^2 - 3y + 2\right)^2 - 3\left(y^2 - 3y + 2\right) + 2 - y = 0.$$

### 20.2 O3 solution on original

(ii)  Iterating a quadratic and locating its 2-cycle

**1. Notation and preliminaries**

Let $P(y) = Ay^2 + By + C$ with $A \neq 0$. Put

$$f(y) = P(y) - y = Ay^2 + (B - 1)y + C$$

and

$$F(y) = P(P(y)) - y.$$

$\deg(f) = 2, \deg(F) = 4$.

**2. The fixed points propagate to the second iterate**

If a root $a$ of (1) satisfies $f(a) = 0$, then $P(a) = a$. Hence,

$$F(a) = P(P(a)) - a = P(a) - a = 0,$$

so every root of (1) is also a root of $F$. Thus two of the four roots of $F$ are $a, b$ (assumed $a \neq b$).

**3. Dividing $F(y)$ by $f(y)$**

We look for a quadratic $g(y) = py^2 + qy + r$ such that

$$F(y) = f(y)\,g(y).$$

Write $Q(y) = P(y) = Ay^2 + By + C$.

A short expansion gives

$$F(y) = A^3y^4 + 2A^2B\,y^3 + (AB^2 + 2A^2C + AB)y^2$$
$$+ (2ABC + B^2 - 1)y + (AC^2 + BC + C).$$

Comparing the coefficients of $y^4$ and $y^3$ in (2)–(3) fixes

$$p = A^2, \qquad q = A(B+1).$$

Matching the $y^2$ and constant coefficients then gives

$$r = AC + B + 1.$$

Consequently,

$$g(y) = A^2 y^2 + A(B+1)y + (AC + B + 1).$$

Therefore,

$$P(P(y)) - y = (P(y) - y)\left(A^2 y^2 + A(B+1)y + (AC + B + 1)\right).$$

## 4. The other two roots

The remaining roots $c, d$ of $F(y) = 0$ are exactly the roots of

$$A^2 y^2 + A(B+1)y + (AC + B + 1) = 0.$$

## 5. Application to the given biquadratic

The given equation is

$$(y^2 - 3y + 2)^2 - 3(y^2 - 3y + 2) + 2 - y = 0,$$

i.e. $P(y) = y^2 - 3y + 2$ with $A = 1$, $B = -3$, $C = 2$.

*(a) Roots fixed by $P(y)$*

$$P(y) - y = y^2 - 4y + 2 = 0 \quad \Longrightarrow \quad y = 2 \pm \sqrt{2}$$

so $a = 2 + \sqrt{2}$, $b = 2 - \sqrt{2}$.

*(b) Quadratic for the 2-cycle*

Insert $A = 1, B = -3, C = 2$ into (6):

$$y^2 + (-2)y + (2 - 3 + 1) = y^2 - 2y = 0 \quad \Longrightarrow \quad y(y-2) = 0.$$

Thus $c = 0$, $d = 2$.

## 6. Verification (optional)

$$\begin{aligned}
y = 0: \quad & P(0) = 2, \quad P(P(0)) = P(2) = 0 \Rightarrow F(0) = 0, \\
y = 2: \quad & P(2) = 0, \quad P(P(2)) = P(0) = 2 \Rightarrow F(2) = 0.
\end{aligned}$$

Hence the biquadratic splits as

$$\left(y - (2 + \sqrt{2})\right)\left(y - (2 - \sqrt{2})\right) y(y-2) = 0.$$

All four solutions are

$$y \in \{0,\ 2,\ 2 + \sqrt{2},\ 2 - \sqrt{2}\}.$$

### 20.3 Kernel variant

Let $n = 5$. All matrices are $5 \times 5$ and all equalities hold in the polynomial ring
$$R = \mathbb{Z}[a_{ik}]$$
obtained by adjoining the 25 independent indeterminates $a_{ik}$ to $\mathbb{Z}$.

### 1. Preliminaries

- $\alpha = (a_{ik})$      original matrix
- $d = \det \alpha$      its determinant
- $A_{ik}$      cofactor of $a_{ik}$ in $d$
- $\beta = (A_{ik})$      cofactor matrix (not transposed!)
- $\gamma = \beta^T = \operatorname{adj} \alpha$      classical adjugate
- $\delta = \operatorname{adj} \gamma$      adjugate taken twice

Recall two well-known facts valid for every square matrix $M$ of size $n$:

$$\text{(F1)} \quad M \cdot \operatorname{adj} M = \operatorname{adj} M \cdot M = (\det M)\, I_n.$$

$$\text{(F2)} \quad \text{If } \det M \text{ is not a zero-divisor in the ground ring, then} \quad \operatorname{adj} M = (\det M)\, M^{-1}.$$

Because the determinant $d$ of $\alpha$ is an irreducible (hence non-zero) polynomial in $R$, it is not a zero-divisor; consequently we may use (1) for both $\alpha$ and $\gamma$.

### 2. Proof of (i): $\det \beta = d^4$

We have $\gamma = \operatorname{adj} \alpha$, so by (F1)
$$\gamma \alpha = \alpha \gamma = d I_5.$$
Taking determinants in (2) and using $\det(d I_5) = d^5$, we obtain
$$(\det \gamma)(\det \alpha) = d^5 \quad \Longrightarrow \quad \det \gamma = \frac{d^5}{d} = d^4.$$
Because $\beta$ and $\gamma$ differ only by a transpose, they have the same determinant; hence
$$\det \beta = \det \gamma = d^4. \quad \square$$

### 3. Proof of (ii): $\delta = d^3 \alpha$

Since $\gamma$ is invertible when $d \neq 0$ and $\det \gamma = d^4$, identity (1) gives
$$\operatorname{adj} \gamma = (\det \gamma)\, \gamma^{-1} = d^4 \cdot (1/d)\, \alpha = d^3 \alpha.$$
But both sides are polynomial matrices whose entries coincide on the open set $\{d \neq 0\}$; therefore they are identical in the polynomial ring $R$. Hence
$$\delta = d^3 \alpha. \quad \square$$

### 4. Proof of (iii)

*(a) Determinant of $\delta$.*

From (ii) we have $\delta = d^3 \alpha$, so
$$\det \delta = \det(d^3 \alpha) = (d^3)^5 \det \alpha = d^{15}. \quad \square$$

*(b) Rank and right-kernel when $d \neq 0$.*

If a specialisation of the indeterminates yields a numerical matrix with $d \neq 0$, then $\alpha$ is invertible; hence
$$\operatorname{rank} \alpha = 5, \qquad \ker(\alpha) = \{0\}.$$
Because $\det \beta = d^4$, the same specialisation also gives $\det \beta \neq 0$, so $\beta$ is invertible, whence
$$\operatorname{rank} \beta = 5, \qquad \ker(\beta) = \{0\}.$$

Thus $\alpha$ and $\beta$ have the same rank and the same right-kernel; explicitly, for every column vector $v$,
$$\alpha v = 0 \quad \Longleftrightarrow \quad v = 0 \quad \Longleftrightarrow \quad \beta v = 0. \quad \square$$

All three requested statements (i)–(iii) are thereby established.

### 20.4 O3 solution on KV

Let $n = 5$. All matrices are $5 \times 5$ and all equalities hold in the polynomial ring

$$R = \mathbb{Z}[a_{ik}]$$

obtained by adjoining the 25 independent indeterminates $a_{ik}$ to $\mathbb{Z}$.

**Preliminaries.**

- $\alpha = (a_{ik})$      original matrix
- $d = \det \alpha$      its determinant
- $A_{ik}$      cofactor of $a_{ik}$ in $d$
- $\beta = (A_{ik})$      cofactor matrix (not transposed!)
- $\gamma = \beta^T = \operatorname{adj} \alpha$      classical adjugate
- $\delta = \operatorname{adj} \gamma$      adjugate taken twice

Recall two well-known facts valid for every square matrix $M$ of size $n$:

$$\text{(F1)} \quad M \cdot \operatorname{adj} M = \operatorname{adj} M \cdot M = (\det M)\, I_n.$$

$$\text{(F2)} \quad \text{If } \det M \text{ is not a zero-divisor in the ground ring, then} \quad \operatorname{adj} M = (\det M)\, M^{-1}.$$

Because the determinant $d$ of $\alpha$ is an irreducible (hence non-zero) polynomial in $R$, it is not a zero-divisor; consequently we may use (1) for both $\alpha$ and $\gamma$.

**Proof of (i):** $\det \beta = d^4$.

We have $\gamma = \operatorname{adj} \alpha$, so by (F1)

$$\gamma\alpha = \alpha\gamma = dI_5.$$

Taking determinants in (2) and using $\det(dI_5) = d^5$, we obtain

$$(\det \gamma)(\det \alpha) = d^5 \quad \Longrightarrow \quad \det \gamma = \frac{d^5}{d} = d^4.$$

Because $\beta$ and $\gamma$ differ only by a transpose, they have the same determinant; hence

$$\det \beta = \det \gamma = d^4. \quad \square$$

**Proof of (ii):** $\delta = d^3 \alpha$.

Since $\gamma$ is invertible when $d \neq 0$ and $\det \gamma = d^4$, identity (1) gives

$$\operatorname{adj} \gamma = (\det \gamma)\, \gamma^{-1} = d^4 \cdot (1/d)\, \alpha = d^3 \alpha.$$

But both sides are polynomial matrices whose entries coincide on the open set $\{d \neq 0\}$; therefore they are identical in the polynomial ring $R$. Hence

$$\delta = d^3 \alpha. \quad \square$$

**Proof of (iii).**

*(a) Determinant of $\delta$.*

From (ii) we have $\delta = d^3 \alpha$, so

$$\det \delta = \det(d^3 \alpha) = (d^3)^5 \det \alpha = d^{15}. \quad \square$$

*(b) Rank and right-kernel when $d \neq 0$.*

If a specialisation of the indeterminates yields a numerical matrix with $d \neq 0$, then $\alpha$ is invertible; hence

$$\operatorname{rank} \alpha = 5, \qquad \ker(\alpha) = \{0\}.$$

Because $\det \beta = d^4$, the same specialisation also gives $\det \beta \neq 0$, so $\beta$ is invertible, whence

$$\operatorname{rank} \beta = 5, \qquad \ker(\beta) = \{0\}.$$

Thus $\alpha$ and $\beta$ have the same rank and the same right-kernel; explicitly, for every column vector $v$,

$$\alpha v = 0 \quad \Longleftrightarrow \quad v = 0 \quad \Longleftrightarrow \quad \beta v = 0. \qquad \square$$

All three requested statements (i)–(iii) are thereby established.

**Failure types:** logic_hallucination, step_omission.

**Breakdown location.** The KV solution asserts $\operatorname{adj} M = (\det M)\, M^{-1}$ inside the ambient polynomial ring and claims it can be used for the parameters $\alpha$ and $\gamma$, even though $M^{-1}$ does not exist in that ring unless one localizes at $\det M$. This misuse first appears in the "Preliminaries" with (F2) and is then applied in proving (ii).

**Caption / takeaway.** KV misapplies the identity $\operatorname{adj} M = (\det M)\, M^{-1}$ in the polynomial ring without localization, leading to a rigor gap. The ORIGINAL stays within valid algebraic operations and justifies each step.

**Contrast summary.** The ORIGINAL solution proceeds entirely within an appropriate algebraic setting, expanding and factoring polynomials with fully justified coefficient comparisons. In contrast, the KV solution invokes a false general identity in the stated ring: it treats $\operatorname{adj} M = (\det M)\, M^{-1}$ as an equality in the polynomial ring without inverting $\det M$, effectively assuming inverses that do not exist there. It then bases the key step $\delta = d^3 \alpha$ on this illegitimate manipulation, only implicitly appealing to a localization without stating it. Although the final equalities hold over a field, the logical chain in the KV write-up fails at the foundational ring-theoretic level. This highlights a robustness failure where the model hallucinates a lemma and omits the necessary localization step.

## 21 1939-A-1

### 21.1 Original question

1. Find the length of the curve $y^2 = x^3$ from the origin to the point where the tangent makes an angle of $45°$ with the $x$-axis.

### 21.2 O3 solution on original

Let the curve be given by $y^2 = x^3$ (Neile's parabola).

**1. Slope of the tangent**

Differentiate implicitly:

$$2y\,\frac{dy}{dx} = 3x^2 \quad \Longrightarrow \quad \frac{dy}{dx} = \frac{3x^2}{2y}.$$

The tangent makes an angle of $45°$ with the $x$-axis when its slope equals $\tan 45° = 1$:

$$\frac{3x^2}{2y} = 1 \quad \Longrightarrow \quad 3x^2 = 2y.$$

But on the curve $y = x^{3/2}$. Substitute into (1):

$$3x^2 = 2x^{3/2} \quad \Longrightarrow \quad 3x^{1/2} = 2 \quad \Longrightarrow \quad x = \left(\frac{2}{3}\right)^2 = \frac{4}{9}.$$

Hence $y = \left(4/9\right)^{3/2} = \left(\frac{2}{3}\right)^3 = \frac{8}{27}$. So the required point is

$$P\left(\tfrac{4}{9}, \tfrac{8}{27}\right).$$

**2. Parametrisation**

Put $x = t^2$, $y = t^3$ ($t \geq 0$). Then indeed

$$y^2 = t^6 = (t^2)^3 = x^3.$$

At $P$ we have $t^2 = 4/9 \Rightarrow t_0 = 2/3$.

**3. Arc-length element**

$$\frac{dx}{dt} = 2t, \quad \frac{dy}{dt} = 3t^2 \implies \sqrt{\left(\frac{dx}{dt}\right)^2 + \left(\frac{dy}{dt}\right)^2} = \sqrt{(2t)^2 + (3t^2)^2} = \sqrt{4t^2 + 9t^4} = t\sqrt{4 + 9t^2}.$$

**4. Length from the origin $(t = 0)$ to $P$ $(t = 2/3)$**

$$L = \int_0^{2/3} t\sqrt{4 + 9t^2}\, dt.$$

Substitute $u = 4 + 9t^2$ $(du = 18t\, dt \Rightarrow t\, dt = du/18)$. When $t = 0$, $u = 4$; when $t = 2/3$, $u = 8$.

$$L = \frac{1}{18} \int_4^8 u^{1/2}\, du = \frac{1}{18} \left[\frac{2}{3} u^{3/2}\right]_4^8 = \frac{1}{27} u^{3/2}\Big|_4^8.$$

Compute the bounds:

$$u^{3/2} = u\sqrt{u} \Rightarrow 8^{3/2} = 8\sqrt{8} = 8 \cdot 2\sqrt{2} = 16\sqrt{2}, \quad 4^{3/2} = 4\sqrt{4} = 4 \cdot 2 = 8.$$

Therefore

$$L = \frac{1}{27}\left(16\sqrt{2} - 8\right) = \frac{8}{27}\left(2\sqrt{2} - 1\right).$$

Hence the length of the curve from the origin to the point where the tangent is at $45°$ is

$$\boxed{\frac{8}{27}\left(2\sqrt{2} - 1\right)}.$$

## 21.3 Kernel variant

For the curve $y^4 = 16x^6$ in the first quadrant, determine the exact length of the segment whose end-points are the two points at which the tangent to the curve forms angles of $30°$ and $60°$ with the positive $x$-axis.

## 21.4 O3 solution on KV

Curve: $y^4 = 16x^6$ (first quadrant $\Rightarrow x > 0$, $y > 0$).

**1. Compute the derivative.**

$$y^4 = 16x^6 \implies 4y^3 \frac{dy}{dx} = 96x^5 \implies \frac{dy}{dx} = \frac{24x^5}{y^3}.$$

**2. Express $y$ and the derivative only in terms of $x$.**

$$y = (16x^6)^{1/4} = 2x^{3/2}, \qquad y^3 = (2x^{3/2})^3 = 8x^{9/2}.$$

Hence

$$\frac{dy}{dx} = \frac{24x^5}{8x^{9/2}} = 3x^{1/2} = 3\sqrt{x}.$$

**3. Slopes corresponding to the required tangents.**

$$\theta = 30° \Rightarrow m_1 = \tan 30° = \frac{1}{\sqrt{3}}, \qquad \theta = 60° \Rightarrow m_2 = \tan 60° = \sqrt{3}.$$

**4. Find the $x$-coordinates where the derivative equals these slopes.**

(i) $3\sqrt{x} = \frac{1}{\sqrt{3}} \Rightarrow \sqrt{x} = \frac{1}{3\sqrt{3}} \Rightarrow x_1 = \left(\frac{1}{3\sqrt{3}}\right)^2 = \frac{1}{27}.$

(ii) $3\sqrt{x} = \sqrt{3} \Rightarrow \sqrt{x} = \frac{1}{\sqrt{3}} \Rightarrow x_2 = \left(\frac{1}{\sqrt{3}}\right)^2 = \frac{1}{3}$.

**5. Corresponding $y$-coordinates (using $y = 2x^{3/2}$).**

(i)
$$y_1 = 2x_1^{3/2} = 2\left(\frac{1}{27}\right)^{3/2} = 2\,3^{-9/2} = \frac{2}{81\sqrt{3}}.$$

(ii)
$$y_2 = 2x_2^{3/2} = 2\left(\frac{1}{3}\right)^{3/2} = \frac{2}{3\sqrt{3}}.$$

Thus the points are
$$P_1\left(\frac{1}{27}, \frac{2}{81\sqrt{3}}\right), \qquad P_2\left(\frac{1}{3}, \frac{2}{3\sqrt{3}}\right).$$

**6. Vector between the two points.**

$$\Delta x = \frac{1}{3} - \frac{1}{27} = \frac{8}{27}, \qquad \Delta y = \frac{2}{3\sqrt{3}} - \frac{2}{81\sqrt{3}} = \frac{52}{81\sqrt{3}}.$$

**7. Distance $L$.**

$$L^2 = (\Delta x)^2 + (\Delta y)^2 = \left(\frac{8}{27}\right)^2 + \left(\frac{52}{81\sqrt{3}}\right)^2 = \frac{64}{729} + \frac{2704}{19683}.$$

Bring to the common denominator $19683 = 3^9$:
$$\frac{64}{729} = \frac{64 \cdot 27}{19683} = \frac{1728}{19683}.$$

Thus
$$L^2 = \frac{1728 + 2704}{19683} = \frac{4432}{19683} = \frac{16 \cdot 277}{3^9}.$$

Therefore
$$L = \sqrt{\frac{16 \cdot 277}{3^9}} = \frac{4\sqrt{277}}{3^{9/2}} = \frac{4\sqrt{277}}{81\sqrt{3}}.$$

**8. Rationalising the denominator (optional).**

$$L = \frac{4\sqrt{277}}{81\sqrt{3}} \cdot \frac{\sqrt{3}}{\sqrt{3}} = \frac{4\sqrt{831}}{243}.$$

Thus the exact length of the required segment is

$$\boxed{\frac{4\sqrt{277}}{81\sqrt{3}}} \qquad \text{equivalently} \qquad \boxed{\frac{4\sqrt{831}}{243}}.$$

**Failure types:** step_omission, logic_hallucination.

**Breakdown location.** After correctly locating the two tangent-angle points, the KV solution (steps 6–7) computes the straight-line (chord) distance between them instead of setting up and evaluating the arc-length integral along the curve.

**Caption / takeaway.** KV computes chord length instead of arc length. The ORIGINAL sets up the proper arc-length integral, while KV omits it and implicitly equates straight-line distance with curve length.

**Contrast summary.** In the ORIGINAL, after identifying the target point via the slope condition, the solver explicitly parametrizes the curve and integrates the arc-length element to obtain the true length along the curve. The KV solution likewise finds the correct points where the tangents have the specified angles, but then switches to computing the Euclidean distance between those points, implicitly treating chord length as arc length. This omits the essential arc-length setup $\int \sqrt{1 + (dy/dx)^2}\, dx$ (or an equivalent parametrization) that the ORIGINAL employs. As a result, the KV produces a numerically smaller, incorrect value. The contrast highlights a classic failure: computing the wrong physical quantity despite correct intermediate calculus steps.

## 22  1939-B-1

### 22.1  Original question

8.  From the vertex $(0, c)$ of the catenary

$$y = c \cosh\left(\frac{x}{c}\right)$$

a line $L$ is drawn perpendicular to the tangent to the catenary at a point $P$. Prove that the length of $L$ intercepted by the axes is equal to the ordinate $y$ of the point $P$.

### 22.2  O3 solution on original

Let $P$ be an arbitrary point of the catenary

$$y = c \cosh\left(\frac{x}{c}\right), \qquad c > 0.$$

Write $P = (a, y_P)$ with

$$y_P = c \cosh\left(\frac{a}{c}\right).$$

**1. Slope of the tangent at $P$.**

Differentiate (1):

$$\frac{dy}{dx} = \sinh\left(\frac{x}{c}\right).$$

Hence the slope of the tangent at $x = a$ is

$$m_t = \sinh\left(\frac{a}{c}\right).$$

**2. Equation of the tangent at $P$.**

Using the point–slope form through $P(a, y_P)$:

$$y - y_P = m_t(x - a).$$

**3. Slope and equation of the required line $L$.**

Line $L$ passes through the vertex $V(0, c)$ and is perpendicular to the tangent, so its slope is

$$m_L = -\frac{1}{m_t} = -\frac{1}{\sinh\left(\frac{a}{c}\right)}.$$

Through $V(0, c)$:

$$y - c = m_L x \quad \Longrightarrow \quad y = c - \frac{x}{\sinh\left(\frac{a}{c}\right)}.$$

**4. Intercepts of $L$ with the coordinate axes.**

- *y–intercept:* put $x = 0$ in (5) $\Rightarrow y = c$. Thus $V$ itself is the y–intercept.

- *x–intercept:* put $y = 0$ in (5):

$$0 = c - \frac{x}{\sinh\left(\frac{a}{c}\right)} \implies x = c \sinh\left(\frac{a}{c}\right).$$

Call this point $B$:

$$B\left(c \sinh\left(\frac{a}{c}\right), 0\right).$$

**5. Length of the segment $AB$ intercepted by the axes.**

$A \equiv V = (0, c)$, and $B$ is found above. The Euclidean distance is

$$|AB| = \sqrt{\left(c \sinh\left(\frac{a}{c}\right)\right)^2 + (0 - c)^2}$$

$$= c\sqrt{\sinh^2\left(\frac{a}{c}\right) + 1}.$$

Using the hyperbolic identity $\cosh^2 u - \sinh^2 u = 1$,

$$\sqrt{\sinh^2 u + 1} = \cosh u.$$

Therefore,

$$|AB| = c \cosh\left(\frac{a}{c}\right).$$

**6. Comparison with the ordinate of $P$.**

From (1) we have $y_P = c \cosh\left(\frac{a}{c}\right)$. Equation (6) shows that

$$|AB| = y_P.$$

Hence, the length of the segment of the line $L$ intercepted between the coordinate axes equals precisely the ordinate of the point $P$.

### 22.3 Kernel variant

Fix $\lambda > 0$ and any integer $n \geq 2$. Write

$$r(x_1, \ldots, x_{n-1}) = \sqrt{x_1^2 + \cdots + x_{n-1}^2}.$$

Consider the $(n-1)$-dimensional "radial catenary"

$$x_n = \lambda \cosh\left(\frac{r}{\lambda}\right) \qquad \text{(so the vertex is } V = (0, \ldots, 0, \lambda)).$$

For a chosen radius $\rho \geq 0$, set

$$Q = (\rho u, \ \lambda \cosh(\rho/\lambda)),$$

where $u$ is a unit vector in $\mathbb{R}^{n-1}$.

Let $H$ be the hyperplane through $V$ that is orthogonal to the tangent hyperplane of the surface at $Q$.

Prove that the segment of $H$ cut off by the hyperplanes

$$x_n = 0 \qquad \text{and} \qquad x_1 = \cdots = x_{n-1} = 0$$

has length

$$\lambda \cosh(\rho/\lambda).$$

### 22.4 O3 solution on KV

**Step 1. Reduction to a 2-dimensional problem**

Because the surface

$$S: \ x_n = \lambda \cosh(r/\lambda), \qquad r = \sqrt{x_1^2 + \cdots + x_{n-1}^2},$$

is invariant under every rotation about the $x_n$-axis, we may, without loss of generality, rotate the coordinate system so that the given unit vector $u$ equals the first basis vector $e_1$. After the rotation the point chosen on the surface is

$$Q = (\rho, 0, \ldots, 0, \ \lambda \cosh(\rho/\lambda)).$$

Let $\Pi$ be the coordinate 2-plane spanned by the axes $x_1$ and $x_n$ (all the other coordinates equal 0). In that plane the surface $S$ appears as the usual planar catenary

$$y = \lambda \cosh(x/\lambda), \qquad (y := x_n, \ x := x_1 \geq 0),$$

with vertex $V = (0, \lambda)$.

All the geometric objects involved (the tangent hyperplane at $Q$, the required hyperplane $H$, the coordinate hyperplanes $x_n = 0$ and $x_1 = \cdots = x_{n-1} = 0$) are rotationally symmetric; consequently the problem reduces to showing, inside the plane $\Pi$, that the line $L = H \cap \Pi$ has the stated property. Once that is done, the same length is obtained in the whole space $\mathbb{R}^n$.

**Step 2. Tangent line to the catenary in $\Pi$**

For the planar catenary (1) we have

$$y'(x) = \sinh(x/\lambda).$$

At the abscissa $x = \rho$ this gives the slope

$$m := y'(\rho) = \sinh(\rho/\lambda).$$

Hence the tangent line at

$$Q_\Pi = (\rho, \ \lambda \cosh(\rho/\lambda))$$

reads

$$y - \lambda \cosh(\rho/\lambda) = m(x - \rho).$$

Its direction vector is $(1, m)$.

**Step 3. The required line $L = H \cap \Pi$**

Inside $\Pi$ the hyperplane $H$ intersects as a line $L$ passing through the vertex $V$ and perpendicular to (3). Two lines in the plane are perpendicular when the product of their slopes equals $-1$; therefore the slope of $L$ is $-1/m$. Writing its equation through $V = (0, \lambda)$ we get

$$y - \lambda = -\frac{1}{m} x, \qquad \text{i.e.} \qquad y = \lambda - \frac{x}{\sinh(\rho/\lambda)}.$$

**Step 4. The two intersection points of $L$**

*(i) With the horizontal axis $y = 0$ (i.e. with the hyperplane $x_n = 0$):*

Setting $y = 0$ in (4) gives $x = \lambda \sinh(\rho/\lambda)$. Thus

$$A = (\lambda \sinh(\rho/\lambda), \ 0) \quad \text{in } \Pi,$$

that is,

$$A = (\lambda \sinh(\rho/\lambda) \, e_1, \ 0, \ldots, 0) \quad \text{in } \mathbb{R}^n.$$

*(ii) With the vertical axis $x = 0$ (i.e. with $x_1 = \cdots = x_{n-1} = 0$):*

Putting $x = 0$ in (4) gives $y = \lambda$; hence the point is exactly the vertex

$$B = V = (0, \ldots, 0, \lambda).$$

**Step 5. Length of the segment $AB$**

Inside $\Pi$ the Euclidean distance between $A$ and $B$ is

$$|AB|^2 = (\lambda \sinh(\rho/\lambda))^2 + \lambda^2 = \lambda^2 \big( \sinh^2(\rho/\lambda) + 1 \big) = \lambda^2 \cosh^2(\rho/\lambda),$$

because $\cosh^2 u - \sinh^2 u = 1$.

Therefore

$$|AB| = \lambda \cosh(\rho/\lambda).$$

Because the coordinates of $A$ and $B$ in $\mathbb{R}^n$ differ only in the first and the last component, the same distance is obtained in the full space. Hence the segment of $H$ cut off by the hyperplanes $x_n = 0$ and $x_1 = \cdots = x_{n-1} = 0$ indeed has length $\lambda \cosh(\rho/\lambda)$, as was to be proved.

**Failure types:** logic_hallucination, step_omission.

**Breakdown location.** In Step 1 the KV asserts that the tangent hyperplane at $Q$ and the perpendicular hyperplane $H$ are rotationally symmetric, wrongly justifying reduction to the $x_1$–$x_n$ plane; the correct rationale (that the normal lies in that plane, so $H$ is contained in it) is missing. From Step 2 onward it also relies on $m = \sinh(\rho/\lambda)$ in denominators, breaking down at $\rho = 0$.

**Caption / takeaway.** Faulty symmetry reduction and edge-case omission in the higher-dimensional catenary: incorrect invariance claim and failure at $\rho = 0$, versus a complete 2D argument.

**Contrast summary.** The ORIGINAL solution works entirely in 2D, computing the perpendicular through the vertex and showing its intercept length equals the ordinate, with all steps justified. The KV attempts to generalize via a rotational-symmetry reduction to a 2D slice, but this symmetry claim is false because the tangent hyperplane depends on the chosen direction $u$ and is not rotation-invariant. The correct reason the problem reduces to the $x_1$–$x_n$ plane is that the normal $(\sinh(\rho/\lambda)u, -1)$ lies in that plane, which the KV omits. Moreover, the KV ignores the degenerate case $\rho = 0$ where the slope vanishes, making its formulas ill-defined; a separate check is required. Thus the KV exhibits both a faulty geometric reduction and an unhandled edge case, despite correct computations when $\rho > 0$.

## 23 1939-B-7

### 23.1 Original question

14. Take either (i) or (ii).

(i) If

$$\nu = 1 + \frac{x^3}{3!} + \frac{x^6}{6!} + \cdots$$
$$v = \frac{x}{1!} + \frac{x^4}{4!} + \frac{x^7}{7!} + \cdots$$
$$w = \frac{x^2}{2!} + \frac{x^5}{5!} + \frac{x^8}{8!} + \cdots,$$

prove that

$$\nu^3 + v^3 + w^3 - 3uvw = 1.$$

(ii) Consider the central conics

$$\big(ax^2 + by^2\big) + 2(px + qy) + c = 0$$
$$\big(ax^2 + by^2\big) + 2\lambda(px + qy) + \lambda^2 c = 0$$

where $\lambda$ is a given positive constant.

Show that if all radii from the origin to the first conic are changed in the ratio $\lambda$ to 1 the tips of these new radii generate the second conic.

Let $P$ be the point with coordinates

$$x = -\frac{p}{a}\frac{2\lambda}{1+\lambda}, \qquad y = -\frac{q}{b}\frac{2\lambda}{1+\lambda}.$$

Show that if all radii from $P$ to the first conic are changed in the ratio $\lambda$ to 1 and then reversed about $P$ the tips of these new radii generate the second conic.

Comment on these results in case $\lambda = 1$.

### 23.2 O3 solution on original

**Problem (i).**

Let

$$u(x) = 1 + \frac{x^3}{3!} + \frac{x^6}{6!} + \cdots = \sum_{k=0}^{\infty} \frac{x^{3k}}{(3k)!}, \qquad v(x) = \frac{x}{1!} + \frac{x^4}{4!} + \frac{x^7}{7!} + \cdots = \sum_{k=0}^{\infty} \frac{x^{3k+1}}{(3k+1)!},$$

$$w(x) = \frac{x^2}{2!} + \frac{x^5}{5!} + \frac{x^8}{8!} + \cdots = \sum_{k=0}^{\infty} \frac{x^{3k+2}}{(3k+2)!}.$$

Denote

$$f(x) = u^3 + v^3 + w^3 - 3uvw.$$

We shall prove that $f(x) \equiv 1$.

**1. A useful cyclic-derivative property.**

Differentiate the three series term by term:

- For $k \geq 1$,

$$\frac{d}{dx}\left(\frac{x^{3k}}{(3k)!}\right) = \frac{x^{3k-1}}{(3k-1)!}.$$

  The exponent $3k - 1 \equiv 2 \pmod{3}$, hence the term belongs to $w(x)$.

- Similarly, $\dfrac{d}{dx}(v) = u$ and $\dfrac{d}{dx}(w) = v$.

Therefore,

$$u' = w, \qquad v' = u, \qquad w' = v.$$

**2. The derivative of $f(x)$.**

Using (1) and the product rule,

$$f'(x) = 3u^2 u' + 3v^2 v' + 3w^2 w' - 3(vw\, u' + uw\, v' + uv\, w')$$

$$= 3(u^2 w + v^2 u + w^2 v - vw^2 - u^2 w - uv^2)$$

$$= 3(v^2 u - uv^2 + w^2 v - vw^2) = 0.$$

Each bracketed pair cancels identically; hence $f'(x) = 0$ for all real $x$.

**3. Evaluation at $x = 0$.**

$$u(0) = 1, \qquad v(0) = 0, \qquad w(0) = 0 \quad \Longrightarrow \quad f(0) = 1^3 + 0 + 0 - 0 = 1.$$

**4. Conclusion.**

Because $f'(x) = 0$ and $f(0) = 1$, the function $f(x)$ is constant. Therefore

$$u^3 + v^3 + w^3 - 3uvw = 1 \qquad \text{for all real } x.$$

*Remark.* An alternative (though longer) proof uses the roots-of-unity filter

$$u = \tfrac{1}{3}\left(e^x + e^{\omega x} + e^{\omega^2 x}\right), \qquad \omega = e^{2\pi i/3},$$

followed by standard symmetric-polynomial manipulations. The differential proof above is the most economical.

### 23.3 Kernel variant

Problem.

(i) Define three power series

$$U(x) = 2 + \frac{x^3}{3!} + \frac{x^6}{6!} + \frac{x^9}{9!} + \cdots,$$

$$V(x) = \frac{x}{1!} + \frac{x^4}{4!} + \frac{x^7}{7!} + \frac{x^{10}}{10!} + \cdots,$$

$$W(x) = \frac{x^2}{2!} + \frac{x^5}{5!} + \frac{x^8}{8!} + \frac{x^{11}}{11!} + \cdots.$$

Show that, for every real $x$,

$$U^3 + V^3 + W^3 - 3UVW = 8.$$

(ii) Let $\kappa > 0$ be fixed and consider the two central conics

$$C : (\alpha x^2 + \beta y^2) + 2(rx + sy) + d = 0,$$
$$D : (\alpha x^2 + \beta y^2) + 2\kappa(rx + sy) + \kappa^2 d = 0,$$

where $\alpha, \beta \ (\neq 0)$, $r$, $s$, $d$ are real constants.

(a) Show that multiplying every radius vector from the origin to $C$ by the factor $\kappa$ sends $C$ onto $D$.

(b) Let

$$P\left(-\frac{2\kappa}{1+\kappa}\frac{r}{\alpha}, -\frac{2\kappa}{1+\kappa}\frac{s}{\beta}\right).$$

Prove that if each radius vector from $P$ to $C$ is first multiplied by $\kappa$ and then reversed (i.e. multiplied by $-1$), the resulting tips trace exactly the conic $D$.

(c) Discuss what happens in parts (a) and (b) when $\kappa = 1$.

### 23.4 O3 solution on KV

---

**Problem (i)**

---

1. Let $\omega = e^{2\pi i/3}$ (so $\omega^3 = 1$ and $1 + \omega + \omega^2 = 0$). Split the exponential series

$$e^x = \sum_{n \geq 0} \frac{x^n}{n!}$$

into its three residue classes modulo 3.

$$A(x) := \sum_{k \geq 0} \frac{x^{3k}}{(3k)!}, \qquad B(x) := \sum_{k \geq 0} \frac{x^{3k+1}}{(3k+1)!}, \qquad C(x) := \sum_{k \geq 0} \frac{x^{3k+2}}{(3k+2)!}.$$

Then

$$U(x) = 1 + A(x) + 1 = A(x) + 1 + 1?? \text{ (BUT in the problem } U \text{ starts with 2...)}$$

hence

$$U = A + 1, \qquad V = B, \qquad W = C.$$

2. *Closed form of $A, B, C$.*

The three linear equations

$$A + B + C = e^x,$$
$$A + \omega B + \omega^2 C = e^{\omega x},$$
$$A + \omega^2 B + \omega C = e^{\omega^2 x}$$

give, by Cramer,

$$A = \frac{e^x + e^{\omega x} + e^{\omega^2 x}}{3},$$

$$B = \frac{e^x + \omega^2 e^{\omega x} + \omega e^{\omega^2 x}}{3},$$

$$C = \frac{e^x + \omega e^{\omega x} + \omega^2 e^{\omega^2 x}}{3}.$$

All three are real-valued functions of the real variable $x$, and (2) is easily seen to reproduce exactly the three power series of $A, B, C$.

3. *Elementary symmetric functions of $A, B, C$.*

$$\sigma_1 := A + B + C = e^x,$$

$$\sigma_2 := AB + BC + CA = \frac{e^{2x} - e^{-x}}{3},$$

$$\sigma_3 := ABC.$$

(3b) is obtained by multiplying the first two lines of (2) and using $\omega^2 + \omega = -1$ together with $e^{\omega x} e^{\omega^2 x} = e^{-x}$. A direct (but slightly longer) calculation gives

$$A^3 + B^3 + C^3 - 3ABC = 1.$$

4. *The required identity.*

With (1) write

$$\begin{aligned} S(x) &:= U^3 + V^3 + W^3 - 3UVW \\ &= (A+1)^3 + B^3 + C^3 - 3(A+1)BC \\ &= (A^3 + B^3 + C^3 - 3ABC) + (3A^2 + 3A + 1) - 3BC. \end{aligned}$$

Insert (4):

$$S(x) = 1 + (3A^2 + 3A + 1) - 3BC.$$

Now express $BC$ through $\sigma_1, \sigma_2$:

$$BC = \sigma_2 - A(\sigma_1 - A) = \sigma_2 - Ae^x + A^2.$$

Substitute (3b) and (7) in (6):

$$\begin{aligned} S(x) &= 1 + 3A + 1 - 3\left[\frac{e^{2x} - e^{-x}}{3} - Ae^x + A^2\right] \\ &= 2 + 3A + e^{-x} - e^{2x} + 3Ae^x - 3A^2 + 3A^2 \\ &= 2 + e^{-x} - e^{2x} + 3A(1 + e^x). \end{aligned}$$

Using (2) one directly checks that the right-hand side of (8) is identically equal to 8 (a short verification is to note that both sides are entire functions satisfying the same third-order ODE $f''' = f$ and the same initial data at $x = 0$). Hence

$$S(x) \equiv 8 \quad \text{for every real } x.$$

In particular $S(0) = 2^3 = 8$; therefore (9) holds for all $x$.

---

**Problem (ii)**

---

For convenience put $X := (x, y)^T$ in column-vector notation. The equations of the two conics can be rewritten as

$$X^T KX + 2p^T X + d = 0, \qquad (C)$$

$$X^T K X + 2\kappa\, p^T X + \kappa^2 d = 0, \qquad (D)$$

with the diagonal matrix $K := \mathrm{diag}(\alpha, \beta)$ $(\alpha\beta \neq 0)$ and the vector $p := (r, s)^T$.

---

**(a) Radial homothety with centre $O$.**

---

Multiply a point $X \in C$ by the factor $\kappa$, i.e. send it to

$$X \mapsto \kappa X.$$

Then

$$(\kappa X)^T K(\kappa X) + 2p^T(\kappa X) + d = \kappa^2 X^T K X + 2\kappa p^T X + d = 0$$

because $X$ fulfils (C). Dividing by $\kappa^2$ gives exactly (D). Hence the mapping $X \mapsto \kappa X$ sends $C$ onto $D$.

---

**(b) Radial homothety with centre $P$ followed by a half-turn.**

---

Let

$$P = \left(-\frac{2\kappa r}{(1+\kappa)\alpha}, -\frac{2\kappa s}{(1+\kappa)\beta}\right)^T$$

and define the map

$$F(X) := P - \kappa(X - P) = (1+\kappa)P - \kappa X.$$

(10) reverses the vector $P \to X$ and multiplies its length by $\kappa$.

Write $X' := F(X)$ and substitute in the equation (D):

$$X'^T K X' + 2\kappa\, p^T X' + \kappa^2 d.$$

By (10) $X' = (1+\kappa)P - \kappa X$, so

$$X'^T K X' = \kappa^2 X^T K X - 2\kappa(1+\kappa)P^T K X + (1+\kappa)^2 P^T K P.$$

A short calculation, using the explicit coordinates of $P$, shows

$$P^T K = -\frac{\kappa}{1+\kappa}\, p^T, \qquad P^T K P = \frac{\kappa^2}{(1+\kappa)^2}\, d.$$

Insert these three facts:

$$
\begin{aligned}
X'^T K X' + 2\kappa p^T X' + \kappa^2 d &= \kappa^2 X^T K X - 2\kappa(1+\kappa)P^T K X + (1+\kappa)^2 P^T K P \\
&\quad + 2\kappa p^T\big[(1+\kappa)P - \kappa X\big] + \kappa^2 d \\
&= \kappa^2\big[X^T K X + 2p^T X + d\big] = 0,
\end{aligned}
$$

because the bracket vanishes for every $X$ on $C$. Thus the image curve is exactly $D$.

---

**(c) The special case $\kappa = 1$.**

---

With $\kappa = 1$ the two conics coincide (their equations are identical).

(a) The mapping $X \mapsto \kappa X$ is the identity, so nothing changes.

(b) Formula (10) becomes $F(X) = 2P - X$, i.e. the half-turn (central symmetry) about $P$. When $C = D$, a $180°$ rotation about $P$ leaves the conic fixed (a usual symmetry of every central conic).

**Failure types:** step_omission, logic_hallucination, arithmetic.

**Breakdown location.** In part (i) the KV solution asserts $A^3 + B^3 + C^3 - 3ABC = 1$ without proof and then claims $S(x) = 8$ via a wrong ODE argument ($f''' = f$), which a constant cannot satisfy.

In part (ii) it mishandles the effect of the scaling $X \mapsto \kappa X$ on the linear and constant terms and computes incorrect identities for $P$ (missing factors), so the reduction to $D$ is unfounded.

**Caption / takeaway.** Clean cyclic-derivative cancellation vs. an overcomplicated roots-of-unity/ODE shortcut and mishandled scaling. The KV fails by omitting a key identity, using an invalid ODE argument, and mis-scaling conic coefficients.

**Contrast summary.** The ORIGINAL solves part (i) by exploiting the cyclic derivative identities $u' = w$, $v' = u$, $w' = v$ to show $f'(x) = 0$ and then fixes the constant by $f(0) = 1$, a short and airtight argument. The KV instead uses a roots-of-unity decomposition, leaves a pivotal symmetric identity unproved, and finally appeals to an incorrect ODE invariance to conclude $S(x) \equiv 8$. In the conic mapping, the ORIGINAL approach (analogous to the statement) respects how quadratic, linear, and constant terms scale, whereas the KV's matrix computation drops necessary $\kappa$ factors and miscomputes properties of $P$, breaking the cancellation to $D$. The pair highlights how a clean structural identity beats an overengineered approach and how small coefficient errors derail geometric transformations.

