# OpenReview forum: "An Investigation of Robustness of LLMs in Mathematical Reasoning: Benchmarking with Mathematically-Equivalent Transformation of Advanced Mathematical Problems"
_ICLR.cc/2026/Conference — Submitted to ICLR 2026_

### Official Review · Reviewer_eLio · 2025-11-01

**Soundness:** 2
**Presentation:** 3
**Contribution:** 3
**Rating:** 2
**Confidence:** 5

**Summary:**

This paper studies the robustness of large language models (LLMs) in mathematical reasoning. The authors propose a Generalization–and–Perturbation (GAP) framework, which tests models on mathematically equivalent but linguistically or parametrically transformed problems. Based on this, they construct PutnamGAP, a large-scale benchmark built from 85 years of Putnam Competition problems, each augmented with five mathematically equivalent variants. Experiments on 18 commercial and open-source models show substantial accuracy drops under surface and parametric transformations, revealing that even advanced models rely heavily on surface cues rather than genuine mathematical understanding.

**Strengths:**

1. Robustness in mathematical reasoning is an increasingly important and underexplored direction. The paper tackles this with a clear motivation and a well-defined experimental setup.

2. The authors introduce five transformation types (four surface-level renamings and one parametric rewrite), providing a systematic way to probe reasoning robustness.

3. The evaluation spans 18 models and demonstrates consistent degradation under mathematically equivalent perturbations, validating the effectiveness of the proposed benchmark.

4. The paper also categorizes error types (symbol confusion, step omission, logic hallucination), offering further insight into model failure patterns.

**Weaknesses:**

1. The experimental analysis is relatively limited and could be enriched by additional studies. (1) It would be useful to include math-specialized models in the evaluation to see how training objectives or dataset composition influence robustness, and to provide insights into how robustness might be improved. (2) The paper could explore whether specific prompting strategies (e.g., instructing models like O1 to pay attention to variable names or perform meta-reasoning) could help defend against these perturbations. (3) It would also be valuable to test whether training on the generated variants can actually improve model robustness. This could show whether the proposed dataset is not only diagnostic but also useful for enhancing reasoning stability.

2. The paper lacks sufficient discussion and comparison with related robustness benchmarks such as GSM-Symbolic, GSM-Plus, and MathCheck, which employ similar perturbation-based methods to test reasoning robustness. A clearer comparison with these works would better situate the contribution of this paper.

**Questions:**

none

---

> ### Author Response · Authors · 2025-11-26
>
> Thank you for your constructive review and highly useful feedback! We appreciate your recognition of many strengths of our paper and made substantial effort to improve the work by following your suggestions.
>
> >The experimental analysis is relatively limited and could be enriched by additional studies.
>
> We agree that our analysis could be further enriched by additional studies, but would also like to point out that our intended primary contribution is the establishment of a general methodology for this kind of diagnostic evaluation and the construction of a new dataset that enables a new kind of analysis of the robustness of LLMs’ capacity of reasoning. The methodology can be potentially used by the research community to construct additional datasets for robustness analysis, and our dataset can also be used by other researchers to further analyze not just the models we analyzed but also many others that may appear in the future. Once again, we do agree that our analysis could be further enriched, and we have done that as described below.
>
> > It would be useful to include math-specialized models in the evaluation to see how training objectives or dataset composition influence robustness, and to provide insights into how robustness might be improved.
>
> Thank you for the excellent suggestion. We actually already tested Deepseek-Prover, a math specialized model in our original paper, which shows that although it has the best robustness compared to other models of similar sizes, the result does not differ significantly. In addition, several recently proposed math-specialized models are not accessible through a public API or released checkpoints, which makes it difficult to include them in a systematic comparison. Following your suggestion, we will further evaluate additional math-specialized models when we could obtain reliable access.

---

> ### Author Response · Authors · 2025-11-26
>
> > The paper could explore whether specific prompting strategies (e.g., instructing models like O1 to pay attention to variable names or perform meta-reasoning) could help defend against these perturbations.
>
> We appreciate the reviewer drawing attention to this Appendix F.5 observation. Our current remark is based on a small pilot where we simply prepended an instruction such as “Replace the variable names with a proper one” to the standard math-solving prompt. On a subset of models and items, this reduced the average accuracy drop on renaming variants to roughly 60% of the original gap (with noticeable variation across models), so we saw a clear but incomplete rebound and therefore only mentioned it briefly in the appendix.
> This indicates that a non-trivial portion of the surface robustness gap is indeed due to front-end symbol encoding and binding: when the model is explicitly asked to canonicalize strange identifiers, it can often reuse essentially the same reasoning that succeeds on the original statement. However, two observations suggest that this is not a complete fix. First, even with this canonicalization hint, a residual drop remains, and our error taxonomy still shows that logical hallucinations dominate over pure symbol-confusion errors. Second, the most damaging surface families (e.g., misleading or garbled names) are precisely those where the identifiers carry spurious or no semantics, so a naive pre-processing layer would itself need to perform non-trivial semantic analysis to decide how to rename them.
> We therefore view prompt- or pre-processing–based canonicalization as a promising way to partially mitigate surface fragility, rather than something that can largely eliminate the robustness gap. In the revision, we plan to expand Appendix F.5 with a small preliminary experiment on a modest held-out subset (on the order of ∼100 problems), comparing the base solving prompt with a few simple canonicalization instructions (e.g., a one-line renaming hint and a symbol-table style instruction) across multiple models. Our goal in this revision is only to quantify how much of the gap can already be reduced by such lightweight prompting; a more comprehensive study of pre-processing and training-based mitigation is left for future work.
>
> > It would also be valuable to test whether training on the generated variants can actually improve model robustness. This could show whether the proposed dataset is not only diagnostic but also useful for enhancing reasoning stability.
>
> We appreciate your insightful suggestion. Indeed, evaluating whether training on GAP-generated variants improves model robustness was part of our planned future work (page  line 476) because such experiments require additional model-training resources and can substantially expand the scope of the paper.

---

> ### Author Response · Authors · 2025-11-26
>
> >The paper lacks sufficient discussion and comparison with related robustness benchmarks such as GSM-Symbolic, GSM-Plus, and MathCheck, which employ similar perturbation-based methods to test reasoning robustness. A clearer comparison with these works would better situate the contribution of this paper.
>
> Indeed, we have essentially followed the same general motivation as these previous studies in the attempt to generate variants of questions to better evaluate the robustness of an LLM. In addition, we have added paragraphs to both the Introduction and Related Work sections that discuss these benchmarks. Our benchmark goes beyond these previous studies to address the question of how to develop a general methodology for generating semantically equivalent variants at scale for complex math questions. Our proposed framework is general and thus can be used by many other researchers to generate variations of complex math problems in the future. Also, GSM8K based variants only include primary school and middle school level **computational** questions with **no proofs**, and, for modern frontier models, it's easy to extract a quantitative relationship from the questions and use calculators to get the answer. Those questions are too easy for recent models and do not have the power to test the difference. Putnam-AXIOM  uses good data source, but it's too small and mixed renaming & parametrical variants, making it difficult to analyze the weakness of models. The following is a preview of a more detailed comparison with the previous work, which we’ll add to the paper.
>
> Recent work has begun to probe mathematical robustness by constructing perturbation-based benchmarks on top of GSM8K and related datasets. MathAttack demonstrates that even simple adversarial paraphrases—constructed to preserve the underlying mathematical structure—can substantially reduce LLM accuracy on a wide range of math word problems (Zhou:2024MathAttack). GSM-Plus augments GSM8K with eight families of adversarial variations per problem, revealing large accuracy drops even for models that nearly solve the original benchmark (Li:2024GSMPlus). GSM-Symbolic builds symbolic templates over GSM8K-style problems and shows that merely changing numeric instantiations or adding logically irrelevant clauses can degrade performance by up to 65% (Mirzadeh:2024GSMSymbolic). MathCheck-GSM further organizes GSM8K-derived problems into a checklist of task and robustness variants to study behavior across multiple evaluation formats (Zhou:2024MathCheck). Beyond GSM8K, GSM8K_MORE uses an ontology of perturbations to generate families of grade-school arithmetic variants (Hong:2025GSMore), while Putnam-AXIOM introduces a smaller set of functional variations for university-level Putnam problems (Gulati:2025PutnamAxiom). These efforts convincingly demonstrate that current LLMs are brittle under controlled perturbations, but they remain largely confined to pre-university word problems or relatively small variation sets, and therefore do not yet provide a large-scale, systematically structured robustness benchmark for competition-level, proof-style mathematics.

---

### Official Review · Reviewer_1ynY · 2025-11-01

**Soundness:** 3
**Presentation:** 3
**Contribution:** 2
**Rating:** 4
**Confidence:** 3

**Summary:**

This paper introduces a new framework, **Generalization-and-Perturbation (GAP)**, to systematically evaluate the mathematical reasoning robustness of Large Language Models (LLMs). The authors argue that existing benchmarks are compromised by data contamination and fail to test for true, generalizable reasoning beyond surface-level pattern matching.

The GAP framework addresses this by "stress-testing" models on **mathematically equivalent** variations of advanced math problems. The paper's contributions are twofold:

1.  **The GAP Methodology:** A novel procedure for generating two classes of problem variants: **surface renames** ($\mathcal{T}_{surf}$), which alter linguistic and symbolic representations (e.g., variable names) , and **parametric rewrites** ($\mathcal{T}_{para}$), which modify the problem's constants and parameters while preserving its core logical structure (termed "Kernel Variant").
2.  **The PutnamGAP Dataset:** A large-scale instantiation of this framework using 1,051 unique Putnam Competition problems from 1938-2024. The authors generate five variants for each original problem (four surface, one kernel), resulting in a 6,306-item benchmark.

Using this benchmark, the authors evaluate 18 prominent commercial and open-source LLMs. The key finding is that all models, including state-of-the-art systems, exhibit a **sharp degradation in performance** when tested on these equivalent variants. For example, the O3 model's accuracy drops by 4.7 percentage points (pp) on surface variants and 12.9 pp on parametric variants, demonstrating that their reasoning capabilities are brittle and sensitive to non-mathematical perturbations.

**Strengths:**

* **Originality:** The paper's originality is high. While robustness testing is not new, the GAP framework's focus on **mathematical equivalence**  is a crucial distinction from prior work on contrast sets or perturbations that change the problem's substance. The specific methodology, distinguishing between surface-level ($\mathcal{T}_{surf}$) and deep-structural ($\mathcal{T}_{para}$) perturbations, provides a novel and insightful way to disentangle different reasoning failures.
* **Quality:** The work is executed with exceptional rigor and quality.
    * **Methodology:** The variant-generation pipelines are sophisticated and well-designed. The use of LLMs to propose replacements for surface renames (Figure 2)  and the complex multi-stage process for "Kernel Variant" synthesis (Figure 3).
    * **Dataset:** The creation of **PutnamGAP** is a contribution on its own. By sourcing 85 years of challenging Putnam problems, the authors have created a high-difficulty, long-lasting benchmark that is unlikely to be "solved" or fully contaminated in the near future.
    * **Metrics:** The authors propose a new, principled **robustness metric $\hat{R}(e,h)$** . This is a strong theoretical contribution, moving beyond simple accuracy ratios to an item-aware, normalized score that better captures catastrophic failures.
    * **Experiments:** The evaluation is comprehensive, covering 18 models , and the auto-grading system is transparently described.
* **Clarity:** The paper is written with outstanding clarity. The motivation is clear , the methodology is explained precisely with helpful diagrams , and the results are presented and analyzed in a way that is easy to follow (Section 5) . The appendices are thorough and provide all necessary details for reproducibility.
* **Significance:** It provides a concrete, actionable methodology for addressing two of the most critical challenges in AI evaluation: **benchmark data leakage**  and **measuring true generalization** vs. "memorized textual surface forms". The GAP framework is general and could be applied to other reasoning domains

**Weaknesses:**

The paper's primary weakness is that it is more descriptive than diagnostic. It excels at *identifying* and *quantifying* the robustness failure but offers limited insight into *why* it occurs or how to fix it.

* **Analysis is Descriptive, Not Diagnostic:** The central finding—that LLM performance drops on perturbed inputs—is, while well-proven, not entirely surprising. The paper stops short of a deep analysis of these failures.
    * The error taxonomy (Section 5.3)  is a good start, but it's still a high-level description. The paper notes that "Logic Hallucination" dominates , but it doesn't explore *why* a simple variable rename (e.g., GS)  or a parameter change (KV)  would cause a *logical* failure, rather than just a "Symbol Confusion" error.
    * The paper misses an opportunity to correlate robustness with model architecture, scale, or training data. Why do some models (gemini-2.5-pro, grok4) suffer a ~15pp drop on KV, while others (gpt-4.1) see a "smaller" ~10pp drop?. A deeper dive into model-specific failures would be highly valuable.
* **Prescriptive Suggestions are Speculative:** The paper's implications section (6.2)  and Appendix F  rightly suggest using the GAP framework for data augmentation and curriculum fine-tuning. However, this is presented as a suggestion. The paper would be much stronger if it included even a small-scale experiment demonstrating this. Proving that fine-tuning on GAP-generated variants *improves* robustness on a held-out test set would "close the loop" and demonstrate the framework's utility for *solving* the problem, not just measuring it.
* **Minor: Potential Grader-Tester Contamination:** The auto-grader relies on an LLM (O3) , which is also used in the variant generation process. While the authors report high human-verified precision (>97%), there is a potential risk that the grader model shares the same sensitivities as the models being tested. For instance, could a "Descriptive_Long_Misleading" (DLM) variant confuse the grader itself? This is not fully addressed.

**Questions:**

1.  **On the Nature of the Findings:** The core finding that performance drops on perturbed data is expected. What do the authors consider the most *surprising* or *non-obvious* result from their experiments? Is it the sheer magnitude of the drop on Kernel Variants (KV)? Or is it the fact that some models *improve* on the "Descriptive Long" (DL) variant, suggesting the original problem phrasing was somehow suboptimal?

2.  **On Deeper Diagnosis:** The paper does a fantastic job quantifying the robustness gap. Can you provide more qualitative insight into *why* these failures occur? For instance, when a model fails on a KV problem, does it fail at the very first step (e.g., setup), or does it successfully follow the original reasoning structure for a while before derailing? A qualitative analysis of *where* in the solution a top model like gemini-2.5-pro (which drops ~15pp on KV ) goes wrong would be invaluable.

3.  **On Closing the Prescriptive Loop:** The paper suggests using GAP for curriculum fine-tuning. This is a key implication. Have the authors run any preliminary experiments to show that training on GAP-generated variants can improve robustness on a held-out set of variants?

4.  **On the Appendix F.5 Observation:** In Appendix F.5, you make a fascinating observation that "if prompted carefully... to rename perturbed variable names back into normal names," models showed "rebounds on performance." This seems to imply the failure is in correctly *encoding* the perturbed symbols, not in the *core reasoning* module. Could you elaborate on this? Does this suggest that the robustness gap for surface variants could be largely mitigated by a simple pre-processing/prompting layer?

5.  **On the Auto-Grader:** The auto-grader also uses an LLM. How can you be confident that the grader is not susceptible to the same perturbations as the models being tested? For example, in your 10% human audit, did you specifically check if the grader's accuracy was lower on the DLM or GS variants compared to the original problems?

---

> ### Author Response · Authors · 2025-11-26
>
> Thank you for your comprehensive review and constructive feedback. We appreciate your recognition of multiple strengths of our work, including the high originality (“the GAP framework's focus on mathematical equivalence is a crucial distinction from prior work”), high quality (“The work is executed with exceptional rigor and quality”),  clarity (“The paper is written with outstanding clarity”), significance (“It provides a concrete, actionable methodology for addressing two of the most critical challenges in AI evaluation”), and generality (“The GAP framework is general and could be applied to other reasoning domains”). We address your concerns and questions as follows.
>
> >The paper's primary weakness is that it is more descriptive than diagnostic. It excels at identifying and quantifying the robustness failure but offers limited insight into why it occurs or how to fix it.
>
> We believe that the whole GAP framework is, in nature, diagnostic since different perturbations can reveal different kinds of weaknesses of a model in reasoning. Indeed, in our experiment results, we have observed that some models are more sensitive to test X, and this is a useful specific insight that may directly help improve the model (“fix it”).
>
> We do agree that it would also be useful to do even deeper analysis of failure than what we have done and analyze the correlation of robustness with model architecture,  but we believe that all those can be left as future work since our intended primary contribution is the GAP framework and the dataset. They provide a solid foundation to enable the research community to use such a methodology and our dataset to potentially perform more in-depth analysis of many models including those that will appear in the future.
>
>
> >Prescriptive Suggestions are Speculative: The paper's implications section (6.2) and Appendix F rightly suggest using the GAP framework for data augmentation and curriculum fine-tuning. However, this is presented as a suggestion. The paper would be much stronger if it included even a small-scale experiment demonstrating this. Proving that fine-tuning on GAP-generated variants improves robustness on a held-out test set would "close the loop" and demonstrate the framework's utility for solving the problem, not just measuring it.
>
> Thank you for the good suggestion. We intended our primary contributions to be about the evaluation methodology, the constructed new dataset, and evaluation results of the major LLM, and thus believe that we can leave the exploration of using our framework to improve a model as future work.  We will add your suggestion to future work.
>
> >Minor: Potential Grader-Tester Contamination: The auto-grader relies on an LLM (O3) , which is also used in the variant generation process. While the authors report high human-verified precision (>97%), there is a potential risk that the grader model shares the same sensitivities as the models being tested. For instance, could a "Descriptive_Long_Misleading" (DLM) variant confuse the grader itself? This is not fully addressed.
>
> We explicitly checked this issue. Cases where DLM (or any variant) confuses the grader are rare, and our reported >97% human-verified precision already counts these errors. DLM does not show a higher misgrading rate than other variant types. In practice, grader-tester contamination has minimal impact.

---

> ### Author Response · Authors · 2025-11-26
>
> >On the Nature of the Findings: The core finding that performance drops on perturbed data is expected. What do the authors consider the most surprising or non-obvious result from their experiments? Is it the sheer magnitude of the drop on Kernel Variants (KV)? Or is it the fact that some models improve on the "Descriptive Long" (DL) variant, suggesting the original problem phrasing was somehow suboptimal?
>
>
>
> We agree that, at a high level, it is not surprising that performance drops under mathematically equivalent perturbations. What we find non-obvious is *how large and structured* these drops are: once we control the solution skeleton via our kernel-variant (KV) pipeline, strong models still show consistent double-digit drops (≈10–15 points) on Putnam-level problems, even though KV instances are provably isomorphic to the originals and pass a strict two-in-a-row verification scheme. This suggests that current models have not internalized parameter-invariant solution patterns, even after saturating easier math benchmarks.
>
> Second, our SD-normalized robustness metric (R_b) shows that this is not just a uniform accuracy shift. Across capable models we consistently observe ($R_{\text{para}} < R_{\text{surf}}$), and the global score ($R_{\text{global}}=\sqrt{R_{\text{surf}}R_{\text{para}}}$) highlights a small but important subset of variants that trigger catastrophic flips which would be almost invisible under a simple accuracy comparison.
>
> Third, the surface variants reveal an asymmetric pattern: benign descriptive renamings (DL) leave accuracy nearly unchanged and can even slightly *improve* some models, whereas misleading or garbled names (DLC/DLM/GS) systematically cost 3–5 points. Combined with our error taxonomy, where logical hallucinations consistently dominate other error types, this indicates that models rely on fragile lexical “semantic anchors” rather than stable abstract proof structures. We will emphasize these non-obvious findings more clearly in the revision.
>
> >On Deeper Diagnosis: The paper does a fantastic job quantifying the robustness gap. Can you provide more qualitative insight into why these failures occur? For instance, when a model fails on a KV problem, does it fail at the very first step (e.g., setup), or does it successfully follow the original reasoning structure for a while before derailing? A qualitative analysis of where in the solution a top model like gemini-2.5-pro (which drops ~15pp on KV ) goes wrong would be invaluable.
>
> We agree that going beyond aggregate numbers to trace-level diagnosis is important. In the current submission we already make a first step via the error taxonomy in §5.3: by clustering grader feedback into four categories (symbol confusion, step omission, arithmetic, logical hallucination) we see a very stable distribution across variants, with logical hallucinations clearly dominating. This suggests that robustness gaps are not caused only by trivial misparsing of the prompt but involve deeper failures of reasoning.
> For kernel variants in particular, our manual inspection of Gemini-2.5-pro on items where the base problem is solved correctly but the KV instance is not shows both failure modes the reviewer hypothesizes. In some failures the model sets up the right objects and follows a reasonable proof outline for several steps before making an invalid manipulation, dropping a case, or omitting a necessary branch. In others the failure is earlier at setup, for example misreading a shifted parameter or ignoring a changed quantifier, which leads it to commit to the wrong line of attack from the start.
> In the revision we will make this diagnosis more explicit by adding a short qualitative subsection, plus a few concrete traces in the appendix, that contrast correct and KV-failure solutions for Gemini-2.5-pro and highlight where they diverge. This qualitative view is meant to complement our existing robustness metrics, rather than introduce new quantitative experiments, and to give readers the kind of “where in the solution does it go wrong?” picture the reviewer is asking for.

---

> ### Author Response · Authors · 2025-11-26
>
> >On Closing the Prescriptive Loop: The paper suggests using GAP for curriculum fine-tuning. This is a key implication. Have the authors run any preliminary experiments to show that training on GAP-generated variants can improve robustness on a held-out set of variants?
>
> We thank the reviewer for highlighting this “closing the loop” direction. In the current submission, we deliberately restrict ourselves to an evaluation study: all reported results are zero-shot, and we never further train the models on GAP-generated variants. The discussion of “curriculum fine-tuning” in §6.2 and in the conclusion was intended as a forward-looking implication of our robustness analysis, not as a claimed contribution of this work. We have not yet run a systematic curriculum study on GAP; to avoid overclaiming, we will revise the text to clearly mark GAP-based training as future work and soften prescriptive wording (e.g., “should become standard practice”) so that it is presented as a promising direction rather than an established result.
>
> >On the Appendix F.5 Observation: In Appendix F.5, you make a fascinating observation that "if prompted carefully... to rename perturbed variable names back into normal names," models showed "rebounds on performance." This seems to imply the failure is in correctly encoding the perturbed symbols, not in the core reasoning module. Could you elaborate on this? Does this suggest that the robustness gap for surface variants could be largely mitigated by a simple pre-processing/prompting layer?
>
> We appreciate the reviewer drawing attention to this Appendix F.5 observation. Our current remark is based on a small pilot where we simply prepended an instruction such as “Replace the variable names with a proper one” to the standard math-solving prompt. On a subset of models and items, this reduced the average accuracy drop on renaming variants to roughly 60% of the original gap (with noticeable variation across models), so we saw a clear but incomplete rebound and therefore only mentioned it briefly in the appendix.
> This indicates that a non-trivial portion of the surface robustness gap is indeed due to front-end symbol encoding and binding: when the model is explicitly asked to canonicalize strange identifiers, it can often reuse essentially the same reasoning that succeeds on the original statement. However, two observations suggest that this is not a complete fix. First, even with this canonicalization hint, a residual drop remains, and our error taxonomy still shows that logical hallucinations dominate over pure symbol-confusion errors. Second, the most damaging surface families (e.g., misleading or garbled names) are precisely those where the identifiers carry spurious or no semantics, so a naive pre-processing layer would itself need to perform non-trivial semantic analysis to decide how to rename them.
> We therefore view prompt- or pre-processing–based canonicalization as a promising way to partially mitigate surface fragility, rather than something that can largely eliminate the robustness gap. In the revision, we plan to expand Appendix F.5 with a small preliminary experiment on a modest held-out subset (on the order of ∼100 problems), comparing the base solving prompt with a few simple canonicalization instructions (e.g., a one-line renaming hint and a symbol-table style instruction) across multiple models. Our goal in this revision is only to quantify how much of the gap can already be reduced by such lightweight prompting; a more comprehensive study of pre-processing and training-based mitigation is left to future work.
>
> >On the Auto-Grader: The auto-grader also uses an LLM. How can you be confident that the grader is not susceptible to the same perturbations as the models being tested? For example, in your 10% human audit, did you specifically check if the grader's accuracy was lower on the DLM or GS variants compared to the original problems?
>
> We explicitly checked this issue. Cases where DLM (or any variant) confuses the grader are rare, and our reported >97% human-verified precision already counts these errors. DLM does not show a higher misgrading rate than other variant types. In practice, grader-tester contamination has minimal impact

---

### Official Review · Reviewer_8Rdo · 2025-11-03

**Soundness:** 3
**Presentation:** 3
**Contribution:** 2
**Rating:** 6
**Confidence:** 4

**Summary:**

Authors introduce  a methodology to evaluate the robustness of LLMs in mathematical reasoning through semantics-preserving perturbations. They call the framework GAP (generalize and perturb). On the surface, the idea is to generate 1) superficial variable renaming patterns and 2) scenario changes, without changing the core semantics or the final answer (/proof steps). Authors instantiate this framework as PutnamGAP, a large-scale benchmark (6.3k) built from 85 years of William Lowell Putnam Competition problems, each expanded into five mathematically equivalent variants. An interesting part is the way the accuracy drops are calculated using "robustness metric", which attempts to cater to the following statsitical issues: i) item-aware or large drops should hurt more than many tiny drops, ii)  scale-free across models and iii) differentiable.
They evaluated 18 LLMs, which suffered significant accuracy drops. They show curriculum learning with randomized symbol identities and numeric parameters help to some extent. They also aim for GAP to be constant source of "leak-proof" test data generation framework.

**Strengths:**

1. The benchmark seems extensive, covering diverse categories, and based on high-level mathematical problems. The important contribution seems to be the evaluation metric. (some concerns below)

2. Evaluation is extensive. Each of the step seems to have been supported by many experiements.

3. Analysis gives us clear insights. The point with curriculum learning is important, but I did not find too much details in the main paper.

**Weaknesses:**

The main motivation is well-known. Other work has tried this in different ways: GSM8k_MORE, GSM-symbolic. Even PUTNAM-AXIOM does this, but not in a scalable way.

The GAP framework seems innovative, though depends a lot of LLMs to do every step. I am unsure how does errors in generation taken care of.

Many important things are in Appendix, which makes the main contributions hard to follow -- like robustness metric details and motivation, curriculum learning training etc.

**Questions:**

L087: Why so many rounds of review? Are you changing some sort of self-verification paramteres? Or are you updating the original question while reviewing?
Given so many rounds, wouldn't it be more feasible to do a manual check?

L105: How dependent are you on $ \pi_i $ ? I would have imagined you would have also used $ \pi_i $ during robustness metric calculation, by evaluating the "path".

While Figures 2 and 3 looks good, it does not help much. I would have rather liked if the question transformations and the data were being shown clearly.

Robustness metric: Why do you need differentiability here? This seems an attempt to combine benchmarking and further training. Is there any reason to combine this? If we use these signals and bake it in the model, doesn't it fail the purpose of being an independent metric. Benchmarking is supposed to be completely independent of the process of model development.

---

> ### Author Response · Authors · 2025-11-26
>
> Thank you for the comprehensive review and constructive feedback. Please find below our response to the main points raised in your review and answers to your questions.
>
> > The main motivation is well-known. Other work has tried this in different ways: GSM8k_MORE, GSM-symbolic. Even PUTNAM-AXIOM does this, but not in a scalable way.
>
> Indeed, we have essentially followed the same general motivation as these previous studies in the attempt to generate variants of questions to better evaluate the robustness of an LLM. In addition, we have added paragraphs to both the Introduction and Related Work sections that discuss these benchmarks. Our benchmark goes beyond these previous studies to address the question of how to develop a general methodology for generating semantically equivalent variants at scale for complex math questions. Our proposed framework is general and thus can be used by many other researchers to generate variations of complex math problems in the future. Also, GSM8K based variants only include primary school and middle school level **computational** questions with **no proofs**, and, for modern frontier models, it's easy to extract a quantitative relationship from the questions and use calculators to get the answer. Those questions are too easy for recent models and do not have the power to test the difference. Putnam-AXIOM  uses good data source, but it's too small and mixed renaming & parametrical variants, making it difficult to analyze the weakness of models. The following is a preview of a more detailed comparison with the previous work, which we’ll add to the paper.
>
> Recent work has begun to probe mathematical robustness by constructing perturbation-based benchmarks on top of GSM8K and related datasets. MathAttack demonstrates that even simple adversarial paraphrases—constructed to preserve the underlying mathematical structure—can substantially reduce LLM accuracy on a wide range of math word problems (Zhou:2024MathAttack). GSM-Plus augments GSM8K with eight families of adversarial variations per problem, revealing large accuracy drops even for models that nearly solve the original benchmark (Li:2024GSMPlus). GSM-Symbolic builds symbolic templates over GSM8K-style problems and shows that merely changing numeric instantiations or adding logically irrelevant clauses can degrade performance by up to 65% (Mirzadeh:2024GSMSymbolic). MathCheck-GSM further organizes GSM8K-derived problems into a checklist of task and robustness variants to study behavior across multiple evaluation formats (Zhou:2024MathCheck). Beyond GSM8K, GSM8K_MORE uses an ontology of perturbations to generate families of grade-school arithmetic variants (Hong:2025GSMore), while Putnam-AXIOM introduces a smaller set of functional variations for university-level Putnam problems (Gulati:2025PutnamAxiom). These efforts convincingly demonstrate that current LLMs are brittle under controlled perturbations, but they remain largely confined to pre-university word problems or relatively small variation sets, and therefore do not yet provide a large-scale, systematically structured robustness benchmark for competition-level, proof-style mathematics.

---

> ### Author Response · Authors · 2025-11-26
>
> >The GAP framework seems innovative, though depends a lot of LLMs to do every step. I am unsure how does errors in generation taken care of.
>
> While theoretically speaking, the GAP framework can also be used by humans to manually generate variants of math problems, it would be a labor-intensive task and can’t be done at scale.  We thus addressed this challenge by leveraging LLMs to generate variants using the GAP framework.
>
> As you rightly pointed out, one concern with using LLMs is the potential errors made by the LLMs. To minimize such errors, we decomposed the whole task into multiple simpler subtasks and ensured that the subtasks delegated to an LLM to perform are all sufficiently simple and the prompts given for a subtask are sufficiently specific; this way,  the probability that an LLM would make a mistake on a task would be minimized. Examples of a subtask include “Exact z-coordinates of the two horizontal planes need only be h apart, not necessarily ±h/2 or symmetric. Please give another possibility.’ , “Illustrative quadratic examples; cone and sphere could be swapped for any solid with cubic term a0 = 0. Please give another possibility.”...… These 2 example questions are asked after the step of structuring the DAG of solutions and mutable slots are found. On other steps, subtasks are equally easy.
>
> How to design a general decomposition algorithm is a challenge. We addressed this challenge by structuring solutions into a general representation with DAGs of steps, finding replaceable slots, and editing the question systematically. Edits are repeated by   traversing on the DAGS, and the  overall solvability is checked by compositing the DAG back into text and checking. Admittedly there could still be errors, but this is the best that we can do at this point. In the future, it is important to further study how to eliminate such errors with symbolic systems.
>
> The decomposition was done by leveraging human expertise to ensure quality control. In this sense, what we used is a human-AI collaboration strategy where we ensure that the task allocated to an LLM is sufficiently simple and clear to be definitely within the capacity of the LLM.

---

> ### Author Response · Authors · 2025-11-26
>
> >Many important things are in Appendix, which makes the main contributions hard to follow -- like robustness metric details and motivation, curriculum learning training etc.
>
> Thank you for pointing this out. We agree that the original 9-page limit con-
> strained our ability to present several multi-line mathematical derivations in
> the main text. In the revision, we are restructuring the paper so that the key
> components of GAP—including the core steps of the pipeline, the derivation of
> the robustness metric, and its main theoretical properties, are moved into the
> main body.
>
> >L087: Why so many rounds of review? Are you changing some sort of self-verification paramteres? Or are you updating the original question while reviewing? Given so many rounds, wouldn't it be more feasible to do a manual check?
>
>
> Thank you for this question - this is a very reasonable concern. The GAP
> dataset contains 6,306 competition-level mathematics problems. Even for highly
> trained undergraduate volunteers, many proof-style answers are subtle or misleading, and manually verifying one problem–variant pair typically takes around 5 minutes. A single full round of manual verification would therefore require well over 500 human hours. Within our institution, it is difficult to allocate that level of expert labor, and the hourly compensation for qualified annotators is typically around $20/h, which makes large-scale manual checking infeasible
> both in cost and in turnaround time.
> That said, we fully agree that running 15 rounds is unnecessary for practical use. In our internal experiments, we observed that the outputs stabilize after roughly 4 rounds. The number 15 reflects the maximum our annotation budget allowed at the time, and we performed these additional rounds out of caution rather than methodological necessity. In the revised manuscript, we will clarify the general recommended protocol (approximately 4 rounds for hard problems such as Putnam-level items), while emphasizing that the appropriate number of rounds should be adapted to task difficulty and model capability. For most future math datasets, especially those substantially easier than Putnam, far fewer rounds would be needed.
> Thank you again for pointing this out. We appreciate the opportunity to make the methodology clearer and more realistic.
>
> >L105: How dependent are you on $\pi_{i}$ ? I would have imagined you would have also used $\pi_{i}$ during robustness metric calculation, by evaluating the "path".
>
> We didn’t use it during the robustness metric calculation because we want the metric to be general since the trajectories of some models like ChatGPT are not accessible.
>
> >While Figures 2 and 3 looks good, it does not help much. I would have rather liked if the question transformations and the data were being shown clearly.
>
> We will include a short, clear, and representative example with differences between variants highlighted. It will be in the appendix since we have no space in the main section.
>
> >Robustness metric: Why do you need differentiability here? This seems an attempt to combine benchmarking and further training. Is there any reason to combine this? If we use these signals and bake it in the model, doesn't it fail the purpose of being an independent metric. Benchmarking is supposed to be completely independent of the process of model development.
>
> Thank you for raising this point. We intended the robustness metric to be (primarily) used for benchmarking and agree that for benchmarking, differentiability is not a required property. Our intent in mentioning differentiability was only as a secondary, ”nice-to-have” property rather than a core requirement. Differentiability is beneficial in the following sense: A differentiable score m(θ, x) (where θ are model parameters and x is an input instance) is locally smooth in θ, which makes it convenient to (i) plot stable robustness curves as a function of training step or hyperparameter, and (ii) optionally use m as an auxiliary reward or regularizer on a separate training distribution. This is analogous to how metrics such as BLEU or ROUGE are used both as standard benchmarks and, in some works, as RL-style rewards on non-benchmark data, without compromising the independence of the held-out test sets they are evaluated on.
> To avoid any confusion, we will explicitly clarify that our robustness metric is designed and used in this paper purely for evaluation, and that its potential to serve as a differentiable signal for future training scenarios is an optional bonus rather than a motivation for the current work.
>
> It is also worth noting that we never back-propagate through the robustness metric, and we do not fine-tune on the GAP benchmark instances, so the metric is used purely as an evaluation signal and the bench- mark remains independent from model development in the usual sense (i.e., no training on the evaluation set).

---

### Author Response · Authors · 2025-11-30
**To AC: A summary of our rebuttal so far**

Dear Area Chair,

We appreciate your time and effort in handling submissions under the current, unusually heavy reviewing workload. To help with your assessment of our paper, we provide a brief summary of (i) the core contributions, (ii) the main points raised by the three reviewers and our responses, and (iii) the concrete improvements we have made/are making in a revised version.

### The core contribution of this paper includes:

"GAP", a Generalization-and-Perturbation framework that automates the generation of mathematically equivalent variants and thereby makes robustness evaluation scalable to competition-level mathematics. Concretely, GAP provides:

- An automated, reusable pipeline that takes a small set of seed problems with human-structured solution DAGs and uses LLMs, under tight verification and decomposition, to generate both surface variants (wording/variable-name changes) and kernel variants (parameter rewrites that preserve the core solution skeleton). This automation is what allows us to scale from a few hand-crafted examples to thousands of high-quality variants.

- PutnamGAP, a 6,306-item benchmark built from 85 years of Putnam competition problems, each expanded into five mathematically equivalent variants. Our benchmark covers problems spanning combinatorics, geometry, algebra, number theory, analysis, ensuring robustness is measured across the full landscape of collegiate problem-solving. This moves robustness testing beyond grade-school word problems to competition-level, proof-style mathematics, while still being generated and checked by a fully specified automated pipeline.

- A principled robustness metric that aggregates accuracy shifts across variants in an item-aware, normalized way and highlights rare catastrophic failures that are invisible to simple average-accuracy comparisons, together with a 18-model evaluation showing substantial and highly structured robustness gaps even for frontier systems.

Overall, GAP provides a large-scale, competition-level, mathematically diverse robustness benchmark, and our perturbation framework reveals statistically significant robustness drops across all evaluated models. By situating competition-level Putnam problems inside structured families of mathematically equivalent variants, GAP and PutnamGAP fill a key gap between accuracy-oriented math benchmarks and generic perturbation suites, and provide a reusable testbed for developing robustness-aware training and evaluation methods for reasoning models.

Reviewers have highlighted these aspects: for example, Reviewer 1 says "Analysis gives us clear insights"; Reviewer 2 describes the work as having “high originality,” “exceptional rigor and quality,” and “outstanding clarity,” and suggests that GAP could be “invaluable” for understanding robustness of mathematical reasoning, while the other reviewers emphasize the extensiveness of the benchmark and the clarity of the setup.

### Here's a summary of the concerns of 3 reviewers and our response:

- Q1: Reviewer 1 and 3 wants a clearer comparison between our contribution (6306 items, generalized from 1051 questions, both extensive on computational questions and proofs) and previous variant-based datasets like GSM8K-Symbolic (8k items, extensive templates only on computational questions, ICLR2025), GSM8K_More (245 items, generalized from 5 questions, computational questions only, ACL2025), Putnam-AXIOM (522 items, computational questions and proofs(mixture), ICML2025).

- A1: We not only introduced PutnamGAP dataset, but also a scalable method on generalizing both computational questions and proofs beyond the template-based computation-only approach of GSM8K-Symbolic. Thus, future researchers can generalize any math dataset they want. Also, our PutnamGAP dataset (i) gathers all 85 years of Putnam questions for the **first time** (OCRed from book). The problems are still challenging to frontier LLMs in 2025 while GSM8K is saturated. (ii) is both larger and harder than most math datasets. (iii) classified different renaming strategies compared with smaller Putnam-AXIOM, making deeper analysis on reasoning possible. You may see more details in our response to reviewers.

- Q2: Reviewer 1 think the presentation of the methodology can be improved by moving some content from appendix to main part to make the paper easier to follow.

- A2: We have 1 more page for camera-ready version! We are going to put core content, including the core steps of the pipeline, the derivation of the robustness metric, and its main theoretical properties, into the main body.

---

> ### Author Response · Authors · 2025-11-30
> **Cont.**
>
> - Q3: Reviewer 1: Why the robustness metric is differentiable? Do you want to train on the metric?
>
> - A3: In this paper, we never back-propagate through the robustness metric, and we do not fine-tune on the GAP benchmark instances, so the metric is used purely as an evaluation signal and the bench- mark remains independent from model development in the usual sense. Differentiability is a desirable bonus of our metric that future researcher may make use of.
>
> - Q4: Reviewer 2 wants deeper diagnosis
>
> - A4: Diagnostic insight and failure analysis. A key concern was that our analysis is “more descriptive than diagnostic,” and that we should more directly address where in the solution trace models fail, especially on kernel variants. We clarified that GAP is inherently diagnostic in that different perturbation families systematically expose different failure modes, and that our existing error taxonomy already shows logical hallucinations dominating over pure symbol-confusion or arithmetic errors across variants. In addition, we inspected cases where a strong model such as gemini-2.5-pro solves the base problem but fails on the kernel variant, and observed both early-setup failures (e.g., misreading shifted parameters or quantifiers and committing to the wrong strategy) and later derailments after initially following a reasonable proof outline. In the revision we plan to add a short qualitative subsection, together with several concrete traces in the appendix, that contrast correct vs. KV-failure solutions and highlight where the reasoning breaks down.
>
> - Q5: Authors mentioned prompting can improve the drop on perturbations. More details?
>
> - A5: In the revision we are expanding Appendix F.5 with a small preliminary experiment (on the order of ~100 problems) that systematically compares the base solving prompt with a few simple canonicalization strategies across multiple models, quantifying how much of the surface gap can be reduced by such lightweight prompting. At the same time, we clarified that the current submission is purely evaluative: all reported results are zero-shot and we never train on GAP-generated variants. Using GAP for curriculum fine-tuning is explicitly framed as future work, rather than as a claimed contribution of this paper. **We already have the results and will report below soon.**
>
> - Q6: LLM-generated variants and auto-grader robustness
>
> - A6: We explained that variant synthesis is decomposed via DAGs of solution steps into simple, local substitution tasks that are well within the capability of current LLMs, and that our internal experiments show that about four rounds of LLM review are sufficient for Putnam-level problems.
>
> - Q7: Authors mentioned curriculum learning on generalized questions and the reviewers want more details.
>
> - A7: We have clarified that evaluating whether training on GAP-generated variants improves model robustness was part of our planned future work (page line 476) because such experiments require additional model-training resources and can substantially expand the scope of the paper.

---

> ### Author Response · Authors · 2025-11-30
> **Cont.**
>
> Overall. We are encouraged that reviewers describe the direction as “fascinating,” the benchmark as potentially “invaluable” for understanding robustness of mathematical reasoning, and the methodology as “rigorous” and “clear.” Our goal in this work is precisely to provide a high-quality, transparently constructed benchmark and framework that make robustness issues at Putnam-level difficulty both visible and measurable for the community.
>
> The list above is a summary, you may find details and a few other questions in our responses or by asking us directly. **Please ask us any questions if you have any concern!**
>
> We hope that the clarifications and planned revisions summarized above help align your assessment of the paper’s scope and contributions with these descriptions, and we thank you again for your careful consideration.

---

> ### Author Response · Authors · 2025-11-30
> **Preliminary results of prompting against perturbations**
>
> We ran a small preliminary experiment on 100 GS (garbled strings) variants. For a strong model, the base solving prompt achieved 48% accuracy (95% CI [0.385, 0.577]), a short canonicalization hint (“rename every confusing identifier to a clear canonical symbol, then solve using the canonical names”) raised this to 58% (95% CI [0.482, 0.672], p=0.0772), while a longer version that also required a detailed “Rename summary” section reached 53% (95% CI [0.433, 0.625], p=0.4414). This suggests that simple canonicalization instructions can partially recover performance on the hardest naming-based surface family. Yet, overly heavy prompting: adding extra bookkeeping and output constraints on top of the core task, would attenuate the gains. Understanding the mechanisms behind this over-prompting effect, and designing more agentic LLM setups that autonomously decide when and how to canonicalize or rewrite the problem context in verifiable domains like mathematics, is an interesting avenue for future work enabled by GAP. Here are the 2 prompts that we added before the baseline problem statement:
>
> ```
> CANONICAL_SOLVER_PREFIX = (
>     "Notice some errors in naming—rename every confusing identifier to a clear canonical symbol, then solve using the canonical names."
> )
> CANONICAL_SOLVER_LONG_PREFIX = (
>     "Notice some errors in naming—rename every confusing identifier to a clear canonical symbol, solve using the canonical names, "
>     "and after you finish append a section titled 'Rename summary:' listing each original name, its canonical replacement, and any remaining naming risks."
> )
> ```

---

### Author Response · Authors · 2025-12-04
**Update: We have updated our paper**

Dear AC,

We have updated the paper together with the full dataset as supplementary material. Thank you for your time.

Best,
Submission 9633 authors.

---

### Meta-Review · Area_Chair_WbZQ · 2026-01-06

**Summary:**

Reviewers agree that the proposed GAP, along with the new benchmark, is a worthy contribution to the community; however, there are several main concerns about the paper.

1. Reviewers 8Rdo and eLio both asked for a detailed comparison with existing robustness benchmarks such as GSM-Symbolic, GSM-Plus, etc.

2. Reviewer 1ynY pointed out that it would be helpful to have more in-depth diagnostic analyses of the failures behind existing models.

3. Reviewers 1ynY and eLio both asked for more concrete, experiment-verified suggestions on how to improve the robustness of the models, including manipulating the prompts and training on the variants proposed in GAP.

4. Reviewer eLio asked for evaluations on math-specialized models.

5. Reviewer 1ynY pointed out that so many details have been pushed to the appendix that the main paper becomes hard to understand.

**Reviewer Concerns:**

1. The comparison with the existing robustness benchmark is well-addressed by the authors' added discussions on related work.

2. While the authors do provide some additional analysis (a small qualitative study section), much of the requested analyses were not provided, because the authors argue that the main intent is to propose the dataset and benchmark that can expose the deficiencies, rather than being diagnostic.

3. The authors partially addressed the prescriptive experiment concern, e.g., by expanding Appendix F.5. Again, the authors argue that more in-depth experiments is out of scope of this paper.

4. The authors pointed out that the current evaluation already included one math model, and that the other math models are not accessible yet.

5. The authors have restructured the paper to move more necessary details back to the main paper.

In short, the only outstanding concerns are more diagnostic and prescriptive analysis. While I agree with the authors that an extensive diagnostic and prescription analysis may be too resource-intensive and should not be considered necessary for a benchmarking paper, I do agree with the reviewers' point that informative, compact diagnostic and prescriptive analyses should be added to this paper.

**Reviewer Scores:**

Reviewer 8Rdo may raise their score to 8 because all the concerns are minor and well-addressed by the rebuttal.

Reviewers 1ynY and eLio are likely to maintain their scores, because the major concerns are deemed out of scope by the authors. However, I believe Reviewer eLio's score is too low given their comments. A more reasonable score is 4.

---

### Decision · Program_Chairs · 2026-01-26

Reject